# On the relationship between the choice of representation and in-context learning

## Abstract

In-context learning (ICL) is the ability of a large language model (LLM) to learn a new task from a few demonstrations presented as part of the context. Past studies have attributed a large portion of the success of ICL to the way these in-context demonstrations are represented, particularly to how labels are represented in classification tasks. On the other hand, observations of the learning capacity of ICL (i.e., the extent to which more in-context demonstrations can lead to higher performance) have been mixed, and ICL is often thought to occur only under specific conditions. The interaction between these two aspects in ICL, representation and learning, has not been studied in depth until now. We hypothesize that they are largely orthogonal of one another, such that the representation of demonstrations determines the baseline accuracy of ICL, while learning from additional demonstrations improves only on top of this baseline. We validate this hypothesis by developing an optimization algorithm that can enumerate a spectrum of possible label sets (representations) varying in semantic relevance. We then perform ICL with varying numbers of in-context demonstrations for each of these label sets. We observed that learning happens regardless of the quality of the label set itself, although its efficiency, measured by the slope of improvement over in-context demonstrations, is conditioned on both the label set quality and the parameter count of the underlying language model. Despite the emergence of learning, the relative quality (accuracy) of the choice of a label set (representation) is largely maintained throughout learning, confirming our hypothesis and implying their orthogonality. Our work reveals a previously underexplored aspect of ICL: the effects of learning from demonstrations and their representations on ICL performance.

## 1 Introduction

LLMs are able to learn a new task from a few examples, an ability known as in-context learning (ICL) (Brown et al., 2020; Dong et al., 2024). A model is prompted with input-output pairs (demonstrations) illustrating the task and then asked to make a prediction for a novel input. The ICL paradigm is appealing as the models appear to learn something new without updating any weights, in contrast with the typical way in which a neural network learns via backpropagation. However, the performance of ICL depends heavily on properties of the given demonstrations (Perez et al., 2021), such as the the distribution of input text, the label space (Min et al., 2022), the number and order of examples (Lu et al., 2021; Liu et al., 2024; Chen et al., 2023; Bertsch et al., 2025), and the overall format of the sequence (Zhao et al., 2021). It remains unclear whether ICL truly constitutes learning, and if so, how learning interacts with elements of the prompt.

Prior work has studied learning and representation in ICL separately, not considering the interaction between the two, which may have led to incomplete conclusions. According to earlier studies, different kinds of in-context learning happen depending on the choice of how labels are represented. In particular, two types of labeling schemes have been studied extensively: gold (or semantically-meaningful) labeling and abstract (or semantically-void) labeling. Pan et al. (2023) found that with an abstract set of labels, smaller models perform similarly regardless of how many demonstrations were presented, while larger models showed increased performance with more demonstrations. This led them to conclude that the emergence of in-context learning depends on the model size. More recently, Kirsanov et al. (2025) observed that LLMs are sensitive to the representation of labels and

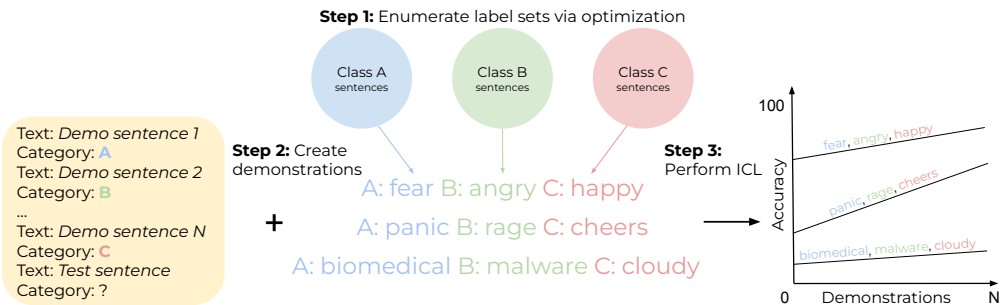

Figure 1: Method overview. Step 1: We develop an optimization algorithm to enumerate a list of possible label sets for a sentiment classification task. Step 2: We label demonstration sentences according to the label sets found. Step 3: We use these demonstrations in ICL tasks and evaluate the performance obtained with each label set on the same set of test sentences.

perform better with gold labels than with abstract labels. In their study, the accuracy improved with an increasing number of demonstrations for both gold and abstract labels, even with a smaller model. Both Min et al. (2022) and Pan et al. (2023) observed that breaking the input-output correspondence while preserving the set of labels had a minimal effect for small models, suggesting that the representation is the sole driver of performance, rather than the demonstration pairings themselves. These findings highlight the need to investigate the interaction between learning and representation in ICL.

In this work, we propose that in classification tasks ICL performance is influenced by two separate components: representation - the choice of class names or labels, and learning - the number of examples presented in context. To quantify the role of representation, we evaluate label sets with varying degrees of semantic relevance to the task. We develop an optimization algorithm to enumerate such label sets. We then use these representations to label input sentences and to create demonstrations for ICL. We conduct experiments on a sentiment classification task: 3-way and 5-way, across three model sizes. For each label set we analyze ICL performance while varying the number of demonstrations. We show an overview of our method in Figure 1.

We found that representation steers learning, although learning typically happens regardless of representation and model size. The *ranking* of representations in terms of accuracy is constant across different number of demonstrations, following the initial order (without any demonstrations). Moreover, the accuracy range attainable with a given representation is largely determined by the zero-shot accuracy. For most label sets, the $N$-shot accuracy generally increases with $N$, although we found that learning efficiency, that is the slope of improvement, depends on the model size. This characterization of the relationship between learning and representation in ICL suggests that it is possible to improve ICL performance by carefully choosing an appropriate label set representation for the task.

## 2 RELATED WORK

There have been a flurry of academic studies on ICL that have revealed its properties and characterized ICL as a new class of learning, since Brown et al. (2020) demonstrated the (surprising) effectiveness of ICL with a large-scale language model. In this section, we list up some of these studies that have shed light on ICL over the past few years.

**Content effects.** Recent studies suggest that LLMs are not fully-abstract reasoners, that is, they do not always learn a function which they can apply to an arbitrary input (Lampinen et al., 2024). Instead, these models show content effects similar to those of humans who reason more accurately about familiar or grounded situations, compared to unfamiliar or abstract ones. McCoy et al. (2024) found that LLM accuracy is influenced by the probability of the task to be performed, the probability of the target output, and the probability of the provided input. The bias towards outputs that have a high prior probability occurs in ICL as well. LLMs do not always identify a unique input-output mapping across the demonstrations, in order to apply it to the test input. They rely instead on the combination of their prior knowledge and presented demonstrations. There are several factors

influencing ICL, such as the order (Lu et al., 2021) and number of demonstrations (Chen et al., 2023), input and output distributions, and the overall format of the prompt (Min et al., 2022). According to these studies, ICL may ignore the task defined by the demonstrations and instead resort to using the prior obtained from pretraining. This implies that ICL may not be considered learning under a strict definition, wherein learning must capture the input-output correspondence in a given training set.

**Learning mechanisms.** Theoretical work has explained ICL as implicit Bayesian inference by training language models from scratch on controlled synthetic data (Xie et al., 2022; Wies et al., 2023; Panwar et al., 2024; Jiang, 2023). Arora et al. (2025) have shown that Bayesian scaling laws are a good fit for the ICL curve. Another line of studies has interpreted ICL as implicitly performing gradient descent (Von Oswald et al., 2023; Ahn et al., 2023) and/or other types of learning algorithms (Akyürek et al., 2023; Garg et al., 2022; Bai et al., 2023; Li et al., 2023). All these mathematical observations encourage us to view ICL as a real learning algorithm and to perform careful empirical investigations to study its properties in real-world settings.

**Pretraining data distribution.** ICL is known to emerge from pretraining when the pretraining data, or its distribution, exhibits a particular set of properties. Chan et al. (2022) found that ICL emerges when data exhibits burstiness (items appear in clusters rather than being uniformly distributed over time) and follows a skewed Zipfian distribution. Raventós et al. (2023) identified a task diversity threshold during pretraining beyond which language models can perform well on unseen ICL regression tasks. Hahn & Goyal (2023) found that ICL arises from generic next-token prediction when the pretraining distribution has a sufficient amounts of compositional structure.

**Prompt optimization.** By deepening our theoretical understanding of the interaction between representation and learning, we can further improve ICL. A common approach to improving LLMs' performance without any extra weight update is via "prompt engineering," that is, by crafting prompts manually. Recent studies introduce prompt optimizers that search over strings to identify high-performing prompts (Yuksekgonul et al., 2025; Zhou et al., 2023; Yang et al., 2024b; Guo et al., 2024; Agrawal et al., 2025). These approaches typically optimize one prompt at a time. For ICL classification tasks, we propose a method to optimize the class names on a separate "labeling" set of sentences, and directly use them as labels in new ICL prompts.

## 3 METHOD

### 3.1 IN-CONTEXT LEARNING FORMULATION

We formulate the goal of an ICL task as solving

$$\arg\max_{y \in \mathcal{C}} p(\tau(y)|x, D_\tau), \tag{1}$$

where $D_\tau = \{(x_n, \tau(y_n))\}_{n=1}^N$ refers to a (small) number of input-output pairs. $\tau(y)$ defines a label set or how we represent each class $y \in \{1, 2, \ldots, \mathcal{C}\}$ as a token in a predefined vocabulary, i.e., $\tau : \{1, 2, \ldots, \mathcal{C}\} \to V$, where $V$ is a vocabulary of unique tokens. $D_\tau$ refers to presenting the dataset $D$ using $\tau$ to encode the classes. By properly formatting $D$, $x$ and $\tau(y)$, LLMs have been found to be able to implicitly learn to predict the correct label associated with a new instance $x$.

Prior work has observed that ICL achieves better performance with gold labels than with abstract labels (Pan et al., 2023). For example, Kirsanov et al. (2025) analyzed a sentiment classification task. The model performed better on an ICL task with gold labels such as {*joy, anger, fear*} than with abstract labels such as {*A, B, C*}, even if the input-output correspondence was the same for both label sets.

While abstract labels lead to worse performance than gold labels, the accuracy increases with more examples for either of the label sets. Based on this observation, and taking into account the content effects revealed by Lampinen et al. (2024), we propose to factor ICL's predictive probability into the product of two probabilities:

$$p(\tau(y)|x, D_\tau) \propto q(\tau(y)|x, D_\tau)p(\tau(y)|x). \tag{2}$$

The first term $q(\tau(y)|x, D)$ corresponds to *learning*, and the second term $p(\tau(y)|x)$ corresponds to *prior* knowledge learned by the language model during pretraining. We assume that the first component, *learning*, is largely invariant to how we represent the classes. In other words,

$$q(\tau(y)|x, D_\tau) \approx q(\tau'(y)|x, D_{\tau'}). \tag{3}$$

On the other hand, the prior knowledge must be sensitive to the choice of $\tau$, as it lacks the context which is presented in the form of in-context demonstrations. Unless $\tau(y)$ is *meaningful* under the pretraining corpus, the language model cannot work with an arbitrary representation of a class *a priori*. That is, it is almost certain that

$$p(\tau(y)|x) \neq p(\tau'(y)|x), \tag{4}$$

for $\tau \neq \tau'$.

In this work, we investigate how the contributions of learning and and prior knowledge are disentangled in ICL. We design a readily actionable way to find a good label map $\tau$ systematically, in order to facilitate this investigation.

## 3.2 CLASS REPRESENTATION OPTIMIZATION

We describe a systematic method to choose a label set $\tau$ that will maximize the performance of ICL across any set of inputs from the same task family. For example, for a sentiment classification task, we can find optimal labels for the classes, and then use these labels as the outputs in ICL demonstrations (input-output pairs) for any other set of inputs.

We assume access to a set of $K$ examples, which we refer to as a labeling set, and knowledge of the class that each example belongs to (how the examples are clustered). The goal is to find, for each class, a name, that is represented by a single token in the vocabulary, that is meaningful under the pretraining corpus. To name $\mathcal{C}$ classes, we want to choose a set of $\mathcal{C}$ tokens from $|V|$ possible tokens in a given vocabulary, $\tau = (l_1, l_2, ...l_\mathcal{C}) \in V^\mathcal{C}$. A good representation map $\tau$ should maximize the probability assigned to the correct class $y^\star$, when represented as $\tau(y^\star)$. We can write this directly as an objective function:

$$\max_{(l_1, l_2, ...l_\mathcal{C}) \in V^\mathcal{C}} \sum_{k=1}^{K} \left( f_\theta(x_k, l_{y_k}) - \log \sum_{c=1}^{\mathcal{C}} \exp(f_\theta(x_k, l_c)) \right), \tag{5}$$

where $x_k$ are the input examples, $y_k \in \{1, 2, ...\mathcal{C}\}$ are the classes they belong to, $l_{y_k} = \tau(y_k)$ is the label assigned to class $y_k$, and $f_\theta$ is the language model's logit. Since the tokens in the label set represent class names and appear after the phrase *"Category:"*, we restrict the vocabulary to tokens that start with the character Ġ (which marks a space and the beginning of a new word).

We optimize this objective via hill climbing, shown in Algorithm 1: we start with an initial random token assignment for each class and iterate the following until no improvements can be made: (1) for each class, try all possible alternative tokens while keeping the rest of class names fixed, (2) evaluate the objective under the current assignment, (3) pick the best token if it improves the overall objective, (4) if there is an improvement, repeat. We run this algorithm ten times while varying random seeds and pick the assignment out of up to ten that maximizes the objective in Equation 5.

As $K$, the number of examples used to find a label assignment, increases it becomes harder to find an assignment for which the labels have high probability for many input sentences. To maximize the objective, that assignment should be generalizable: class names should be meaningful for other possible inputs. Thus, as $K$ increases, we expect the semantics of the labels to be closer to those of gold labels. Equivalently, those labels' zero-shot accuracy for new inputs would be higher with larger $K$. By exploiting the dependence of quality on $K$, we obtain a diverse set of label groups that vary in their semantic relevance to the given classification task.

## 4 EXPERIMENTAL SETUP

We conduct a series of experiments to test the hypothesis that learning and representations are largely disentangled in ICL. First, we want to test whether learning emerges regardless of the choice of

---

**Algorithm 1** Hill Climbing for Token Assignment Optimization

---

**Require:** Initial token assignment for each class
**Require:** Set of candidate tokens, training sentences with labels
**Ensure:** Optimized token assignments
 1: **function** HILLCLIMB($initial\_assignments$)
 2:     $assignments \leftarrow initial\_assignments$
 3:     $objective \leftarrow$ CALCULATEOBJECTIVE($assignments$)
 4:     **repeat**
 5:         $improved \leftarrow$ False
 6:         **for** each $class$ in classes **do**
 7:             $candidates \leftarrow$ all tokens except current token for $class$
 8:             **for** each $token$ in $candidates$ **do**
 9:                 Compute total objective value assigning current token to this class, Eq. 5
10:             **end for**
11:             $best\_token \leftarrow$ token with highest objective
12:             **if** $best\_token$ improves current objective **then**
13:                 $assignments[class] \leftarrow best\_token$
14:                 Update $objective$
15:                 $improved \leftarrow$ True
16:                 **break**                                                     ▷ Try next class
17:             **end if**
18:         **end for**
19:     **until** not $improved$ or max iterations reached
20:     **return** $assignments, objective$
21: **end function**

---

label representation. For this to be true, for any label set, the $N$-shot accuracy should be increasing with $N$. Second, we want to see how representations influence the learning trajectory. For this, we look at how the $N$-shot accuracy relates to the zero-shot accuracy (for the test input) across the different label representations. We conduct experiments with three different size open-weight models: Llama 3.2 1B, Llama 3.1 8B, Llama 3.1 70B Instruct (Grattafiori et al., 2024). We first apply the optimization Algorithm 1 to obtain a series of label sets with varying quality for a classification task. Then, we sample demonstrations and name the outputs according to the label set. We prompt a model with the relabeled and concatenated demonstrations to evaluate the ICL performance on these new inputs.

**Data and prompting.** We use a synthetic sentiment classification dataset from Kirsanov et al. (2025), which contains 1,000 sentences split equally among 5 classes for 5-way classification. We also use a subset of 600 sentences covering only 3 of the classes for 3-way classification. We split the dataset into a labeling set (25%), a demonstration set (25%), and a test set (50%). The labeling set is used to enumerate class name assignments, the demonstration set is used for the support examples for ICL, and the test set is used for the query inputs in ICL. For each $N$-shot classification task, the task is presented in a minimal format with no explicit instructions, only $N$ demonstrations and a query sentence.

**Label sets.** We evaluate different label sets in ICL. These label sets do not break the original input-output correspondence and only replace the original label names, i.e. the assignment of the classes remains the same. Each label set is obtained by optimizing Equation 5 using $K \in \{10, 20, ...100\}$ examples. We show the label sets found with each of the three models in Appendix A Table 1 for 3-way classification and Table 2 for 5-way classification. The examples used for finding a label set are the same for each fixed $K$ across all model sizes. Some of the $K$ values (adjacent ones) resulted in the same label set.

We illustrate a few of the label sets obtained for 3-way classification with the 70B model. Naturally, using a small $K = 10$ leads to overfitting on labels that have a high zero-shot probability only for those labeling examples. This yielded random words as labels such as {*biomedical, malware, cloudy*}. With a small $K$, we cannot find label sets that appear relevant for a sentiment classification task. For a medium value of $K = 40$, the labels obtained are more general {*panic, rage, Cheers*}, a much better fit for the task. While these labels are clearly descriptive, they are slightly odd choices

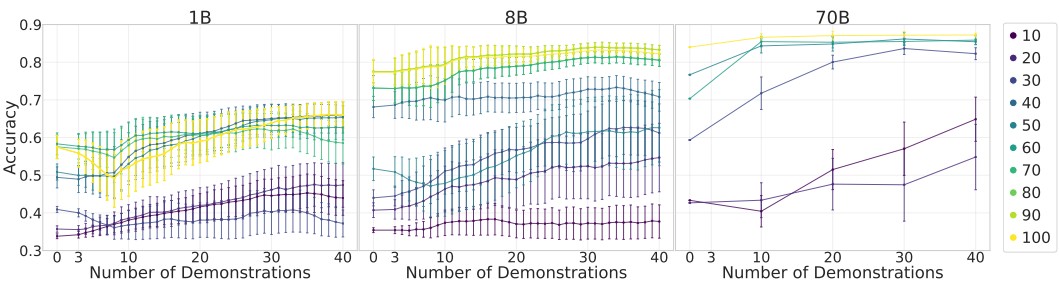

(a) 3-way classification. 1B: $K \in \{80, 90, 100\}$ examples resulted in the same set; 8B: $K \in \{60, 70\}$ and $K \in \{80, 90\}$ same set; 70B: $K \in \{40, 50\}$ and $K \in \{70, 80, 90, 100\}$ same set

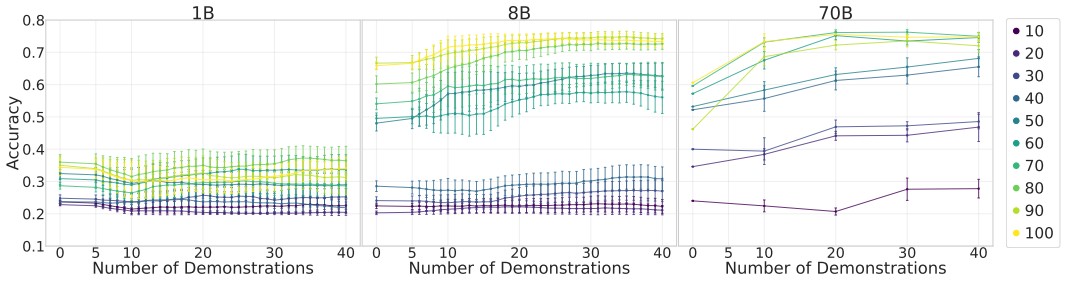

(b) 5-way classification. 70B: $K \in \{70, 80\}$ same set

Figure 2: Accuracy vs. number of demonstrations across model sizes for (a) 3-class and (b) 5-class settings. The curves were smoothed with a window size of 10, with error bars showing 95% CI over 10 runs. The legend shows the number of labeling examples $K$ used to fit the label set. Different $K$ values may result in the same label sets. For these sets, the color shown is that of the higher $K$.

for class names. Finally, using a large $K = 70$ leads to natural category names for a sentiment classification task such as {*fear, angry, happy*}.

The label sets obtained with the same $K$ value vary with different models. For instance, for $K = 100$, the 1B model found {*spectacle, dance, condolences, peril, pissed*}, the 8B model found {*surprising, joyful, sorrow, fears, anger*}, and the 70B model found {*surprise, happy, sad, anxious, ang*}. In general, the label sets found by larger models appear to be more semantically meaningful.

**In-context learning.** We sample $N \in \{0, 1, ...40\}$ examples from the demonstration set and name them according to one of the label sets previously obtained. For the 70B model, we only ran experiments with $N \in \{0, 10, 20, 30, 40\}$ due to compute limitations. For the 1B and 8B models, we ran experiments with $N$ up to 100, as shown in Appendix B. We create demonstrations with a given label set by using that set to label the inputs in a context, and preserve the original input-output mapping from the dataset. These input-output pairs are concatenated, and, together with a query, are given as a prompt to a model. The model then predicts the class for a novel input selected from the test set, which has not been shown in any of the demonstrations and was not used to compute the label sets. We report the average accuracy for the test set, over 10 runs, in which the inputs of the demonstrations are resampled every time.

## 5 RESULTS

Figure 2 shows the accuracy vs. number of demonstrations in ICL tasks with different label sets for the Llama 1B, 8B, and 70B models, for 3-way (Figure 2a) and 5-way classification (Figure 2b) for the sentiment analysis task. In Appendix C we show results with Mistral-7B-v0.3 (Jiang et al., 2023) and Qwen2.5-7B (Yang et al., 2024a; Team, 2024). In Appendix D we also use a different question classification dataset, TREC (Hovy et al., 2001; Li & Roth, 2002). Across all experimental conditions, we observe that the accuracy is generally increasing with the number of demonstrations. There are exceptions, such as when the label set found has a very small zero-shot test accuracy, most

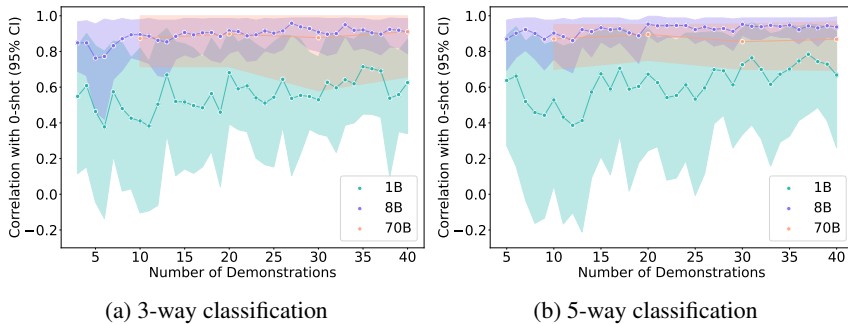

(a) 3-way classification        (b) 5-way classification

Figure 3: Ranking correlation coefficient between the zero-shot accuracy and the $N$-shot accuracy vs. $N$ number of demonstrations. $N \in \{\text{num classes}, ...40\}$ for 1B and 8B models, $N \in \{10, 20, 30, 40\}$ for 70B model. The CI are computed over 1000 bootstrapping samples from 10 runs per N-shot accuracy. **The order of label sets in terms of quality stays consistent across $N$-shot experiments.**

curves stay flat, especially for the harder task of 5-way classification. The zero-shot accuracies span a a wide range from chance to ceiling: 33% to 87% for 3-way classification and 20% to 76% for 5-way classification. The representations with a lower zero-shot accuracy typically resulted from optimization on a small $K$ labeling examples, while those with a high zero-shot accuracy resulted from a larger $K$. The ordering of the label sets as determined by their zero-shot accuracy generally stays constant across $N$-shot tasks, suggesting a consistent ranking of label sets in terms of ICL performance, regardless of the number of demonstrations.

## 5.1 Role of representation in ICL

**Consistent label set ranking.** The $N$-shot accuracy of an ICL task using a label set depends on the zero-shot accuracy with that label set: the $N$-shot accuracy is typically higher for label sets with higher zero-shot accuracy and can only grow up to a limit. This is consistent across label sets. ICL performs better if the label set is meaningful under the pretraining corpus. We observe that for each $N$-shot classification task, the accuracies for ICL with different label sets are ordered according to their initial zero-shot accuracy. We compute the ranking correlation between the zero-shot accuracies and the $N$-shot accuracies (of all the label sets) with $N \in \{\text{num classes}, ...40\}$ for 1B and 8B models, $N \in \{10, 20, 30, 40\}$ for 70B model. We find that the correlations are indeed high across all model sizes, for both 3-way and 5-way classification (see Figure 3), although there is a lot of variance for the 1B model.

**Representation limits the accuracy range.** If the zero-shot accuracy of a given label set is low, it is very difficult for ICL to reach a high accuracy regardless of how many demonstrations are used. Reaching a high accuracy with a low zero-shot accuracy label set might require a very large number of demonstrations. Most of the curves appear to increase more slowly around 40 demonstrations, indicating a possible upper bound. The chosen label set thus largely determines the range of accuracies attainable with that representation. However, there are exceptions where the accuracy has not yet plateaued with 40 demonstrations (see Figure 2a 70B model, $K = 10$), suggesting that it is possible to overcome the limits of the representation with a large number of demonstrations and a larger model. Our findings indicate that the choice of representation is an essential factor when studying ICL and the role of demonstrations, and they shed light on some earlier findings. For example, Pan et al. (2023) found that an abstract label set underperformed random allocation of the gold labels to the inputs of the demonstrations, and claimed that this meant that the models could not truly learn the task, but rather relied on their priors. We instead attribute their finding to the fact that the abstract label set has a much lower zero-shot accuracy than a gold label set, and the accuracy increase from learning from additional demonstrations was insufficient to overcome the baseline limitation, which is typically the case for smaller models.

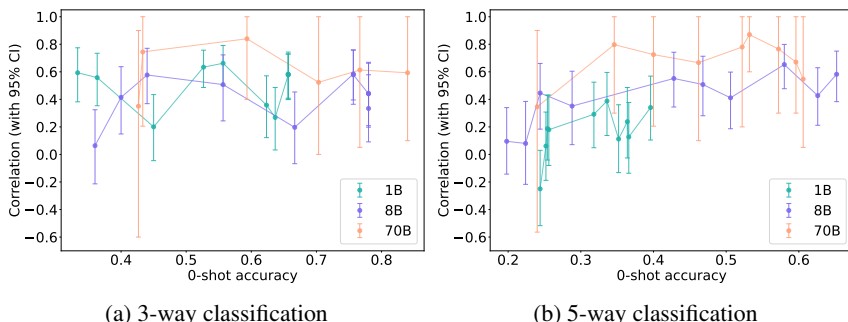

(a) 3-way classification          (b) 5-way classification

Figure 4: **Evaluation of learning curves for label sets obtained with different $K$ labeling examples.** Ranking correlation coefficient between $N$ and $N$-shot accuracy vs. zero-shot accuracy for each curve. $N \in \{$num classes$, ...40\}$ for 1B and 8B models, $N \in \{10, 20, 30, 40\}$ for 70B model. Higher correlation indicates that the accuracy for that curve is often strictly increasing with $N$ (steeper curve), while lower accuracy indicates that the accuracy can be plateauing or decreasing on some intervals (flatter curve). The CI are computed over 1000 bootstrapping samples from 10 runs per N-shot accuracy.

## 5.2 WHEN DOES ICL LEARN?

**Learning almost always happens.** We observe that if the zero-shot accuracy is above some threshold, the curves are always increasing regardless of the model size. For the 3-way classification task (Figure 2a), the threshold zero-shot accuracy is very low (33%, chance level), and all curves increase monotonically. For the 5-way classification (Figure 2b), the threshold is higher (40%, double the chance accuracy). Smaller language models can only in-context learn under specific conditions (Schick & Schütze, 2021), and sometimes not at all. We also observe that this is the case, especially for the harder task of 5-way classification, when labels that are not sufficiently semantically meaningful are used. It appears that with a sufficiently good representation, all models, regardless of size, are able to benefit (to different extents) from more demonstrations.

**Model size influences the learning rate.** From Figure 2 we observe that most learning curves are increasing. In Figure 4 we show that the slope depends on model size and zero-shot accuracy. The larger 70B model is more efficient; it makes more use of fewer examples and thus exhibits steeper curves (such as Figure 2a, 70B model, $K = 30$). The $N$-shot accuracies for this curve are highly correlated with $N$ (see Figure 4a orange curve, zero-shot accuracy 59%). With representations of a similar zero-shot accuracy (40%-60% range), the smaller models can also learn, but their curves increase more slowly (and thus have a lower correlation between $N$ and $N$-shot accuracy), suggesting that it would take many more demonstrations to attain the same accuracy that the 70B model achieves with 20-30 demonstrations.

**Learning is conditioned by representation.** Most of the learning curves typically increase, but there is a lot of variance in how much ICL improves with more demonstrations. The increase between the minimum accuracy (zero-shot) and the maximum accuracy (40-shot) ranges from 0% up to 25%. We observe that the representations fall into three categories: small, medium, and high zero-shot accuracy. The small, zero-shot accuracy representations are usually found with a small $K$ number of labeling examples and are not intuitive or appropriate names for the task. This type of label set makes the task challenging: the model may have to infer the true nature of the task (possibly by inferring more suitable class names) and then map the unintuitive labels onto them. It is not always apparent from the sentences that they illustrate a sentiment classification task. For example, a sentence like "*In the upcoming season, I'll be in the zone every time I step onto the court.*" labeled with "*cloudy*," might distract the model from the clustering of sentences into appropriate classes. Typically for representations like this, the models start with near-chance zero-shot accuracy, and the accuracy increases only very little regardless of how many demonstrations are presented (e.g. Figure 2a 8B model, $K = 10$ and Figure 2b 8B model, $K \in \{10, 20\}$). The representations with a medium (40%–60%) zero-shot accuracy benefit the most from demonstrations. They can get 15%–25% improvement from the baseline by seeing demonstrations. These labels are sufficiently

suggestive of the task {*medically, offending, celebrations*} that the model can eventually determine the mapping.

The last group of representations consists of the high zero-shot accuracy representations, those that match or are very close to gold labels. These label sets are already close to the ceiling accuracy possible for each model size. In Figure 2a, for all the models, the curve corresponding to the highest zero-shot accuracy only obtains a 3–5% increase from the baseline before it plateaus. In this group, we observed one exception. In Figure 2a, for 3-way classification with a 1B model, the curve corresponding to $K \in \{80, 90, 100\}$ initially decreases before increasing. One of the labels in this set is the translation of the word *danger* in Nepali. The ICL task may be harder because it requires multilingual reasoning, which can involve translation as a first step before figuring out the input-output mapping. In the zero-shot case, there is only one test sentence, and no labels appear in the prompt. The model simply predicts the high probability token, even if it is in a different language than the input, since it does not "know" that the other labels are in a different language. It appears that for $N < 9$ examples, the model is confused and thus the accuracy decreases. This might happen since as soon as the demonstrations are shown ($N \geq 3$), all three labels appear, so one of them being in a different language than the rest adds the complexity of translation to the original classification task. With enough demonstrations, the model recovers and achieves a high accuracy toward $N = 40$ demonstrations, as expected for the corresponding zero-shot accuracy.

## 6 CONCLUSION

The success of ICL has previously been attributed to how the in-context demonstrations are represented, and prior work has questioned whether true learning is, in fact, happening (Perez et al., 2021; Min et al., 2022). Previous observations show that ICL performance improves with the number of demonstrations for both gold and abstract labels (Kirsanov et al., 2025), with gold labels consistently outperforming abstract ones. Based on this, we hypothesized that the choice of representation influences the learning trajectory in ICL. We developed an algorithm to enumerate a spectrum of label representations varying in semantic relevance and tested the performance of these label sets in ICL. We found that the representation of demonstrations determines the baseline accuracy of ICL, as measured by zero-shot performance. The relative quality of the label sets is consistent across demonstrations, and follows the order determined by the baseline accuracies. Furthermore, this baseline typically limits the range of attainable accuracies. It is possible to overcome the limits of the representation, but only with a large number of demonstrations and larger models. The efficiency of learning, measured as the slope of improvement over in-context demonstrations, is influenced both by the quality of representation and model size. Representations with a medium zero-shot accuracy typically benefit the most from seeing more demonstrations and have a higher slope, and larger models can learn faster. In summary, our work reveals the relationship between number of demonstrations and how they are represented on ICL performance, and highlights the importance of considering the representation when studying properties of in-context learning from demonstrations.

Our findings on the interaction between learning and representation in LLMs closely reflect what we know of more conventional neural network learning. The search for high-performing prompts for LLMs is in spirit similar to hyperparameter search (Bengio & LeCun, 2007; Liu et al., 2019) for neural network classifiers that learn via backpropagation. Perez et al. (2021) found that good prompts are effective because they are chosen using large validation sets. The prompts influence the model behavior similarly to how a choice of initialization influences neural network training. In particular, the choice of label representation in ICL is analogous to the feature selection for the inputs of a neural network classifier. The different choices of representation determine the learning trajectory in both cases: for LLMs, a high quality representation leads to a high zero-shot accuracy and faster convergence; for neural network classifiers, a good set of features can lead to efficient learning (LeCun et al., 2012).

Our framework has a few limitations. First, we assume that the labels are limited to one token, which might not be the case in more complex ICL tasks. We chose to only explore this option since the search space for finding label sets with our algorithm grows exponentially with the number of tokens. Future work should investigate the case with multi-token labels. Second, we focus only on how labels are represented. Variations in input representations or prompt format may also

influence learning. For instance, choosing inputs with high priors under the model could improve efficiency (Ceballos-Arroyo et al., 2024). Although we treated inputs as fixed, real-world datasets often allow for choosing among many possible inputs. Thus, future work should examine how the inputs and prompt structure affect learning. Third, the in-context learning setting we studied is restricted to one round of prompting. It would be interesting to study the how multi-turn interactions influence ICL in cases where the demonstrations are presented in an online manner, possibly together with extra information in each round (Lee et al., 2023).

Beyond in-context learning, LLMs have shown high performance on complex reasoning tasks, such as programming and mathematical problem solving (Guo et al., 2025; Ruis et al., 2025). Our study also has potential implications about the role of representation in such reasoning. The finding that the representation determines both a baseline accuracy and the efficiency of in-context learning suggests that LLMs already have useful priors, but in order to make the most use of them, we need to present the task in an appropriate manner. Extending these findings about ICL to more complex reasoning tasks could offer a more nuanced understanding about memorization vs. reasoning in LLMs (Bowen et al., 2024; Jin et al., 2025; Salido et al., 2025). Moreover, our findings could explain LLM reasoning failures when changing parameters of an original problem such as document length or the number of variables in a math problem (Malek et al., 2025). Such changes in the prompt, despite attempting to preserve the fundamental difficulty of a problem, result in a significant change in the representation, which lowers the baseline accuracy.

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

# A  LIST OF LABEL SETS

We show the label sets obtained from Algorithm 1 on the sentiment analysis task.

| K | 1B | 8B | 70B |
|---|---|---|---|
| 10 | Nutrition, Giz Legends | Gluten, Laptop clouds | biomedical, malware cloudy |
| 20 | diabetes, Hacker Presbyterian | Diabetes, Revenge spirit | fitness, computer joyful |
| 30 | overweight, annoy scholarships | FDA, console celebration | Obesity, rage celebration |
| 40 | medically, offending celebrating | fearful, malicious celebration | panic, rage Cheers |
| 50 | medically, offending celebrations | digestive, insulting accomplishments | panic, rage Cheers |
| 60 | panicked, offending celebrations | fears, insults joyful | worry, complain celebration |
| 70 | hazardous, offending celebrations | fears, insults joyful | fear, angry happy |
| 80 | à¤kà¤¤à¤° (*danger*, Nepali), offending celebrations | fears, complaints joyful | fear, angry happy |
| 90 | à¤kà¤¤à¤° (*danger*, Nepali), offending celebrations | fears, complaints joyful | fear, angry happy |
| 100 | à¤kà¤¤à¤° (*danger*, Nepali), offending celebrations | fears, complaint joyful | fear, angry happy |

Table 1: Label sets obtained from running Algorithm 1 on $K$ labeling examples for 3-way classification. The gold labels are "fear, anger, joy."

| K | 1B | 8B | 70B |
|---|---|---|---|
| 10 | movie, Musik Causes, Roller NRL | theater, COLOR HEALTH, ride Offensive | Marvel, MUSIC HEALTH, roller veh |
| 20 | witches, audition bere, adip Messi | cinema, Broadcasting Deng, nut Rugby | Magical, positive loss, Dietary Baseball |
| 30 | trick, Dresses bere, hysteria Messi | surprising MÃ©d (*Med*, French) ÑġÐ¾Ð½ (*dream*, Russian) snack, soccer | surprise, celebration tragedy, amused ì˦íɪ¬ì¸ł (*sports*, Korean) |
| 40 | puzzle, Ventures à¤h (*A*, Marathi), carniv Penalty | surprising ÑÂÐµÐ» (*tel*, Russian) resignation, aliment soccer | surprising, positives heartbreaking Ð¿Ð¸Ñī (*shout*, Bulgarian) frustrated |
| 50 | spectacle, talent mourn, endanger offense | surprised, baÅŁarÄ± (*success*, Turkish) sadness, xen, Rage | surprising, positives heartbreaking, Brussels frustrated |
| 60 | spectacle, production mourn, peril offense | amazed, Ð½Ð°ÑĥÐº (*science*, Ukrainian) mourn, scare, brawl | surprising, positives heartbreaking, nerv agg |
| 70 | spectacle, productions mourn, peril racket | surprising, íh (?) sorrow, terror hostile | surprise, pleasant sorrow, fears rage |
| 80 | spectacle, dance mourn, peril criticizing | surprising, celebrates condolences, terror rage | surprise, pleasant sorrow, fears rage |
| 90 | magician, dancer mourning, risking wrath | surprising, joyful sorrow, fears rage | surprise, Lift broken, fears rage |
| 100 | spectacle, dance condolences, peril pissed | surprising, joyful sorrow, fears anger | surprise, happy sad, anxious ang |

Table 2: Label sets obtained from from running Algorithm 1 on $K$ labeling examples for 5-way classification. The gold labels are "surprise, joy, sadness, fear, anger."

# B    LEARNING CURVES

We show the full raw (unsmoothed) learning curves for up to 100 demonstrations for Llama models for 3-way classification in Figure 5 and for 5-way classification in Figure 6 for the sentiment analysis task.

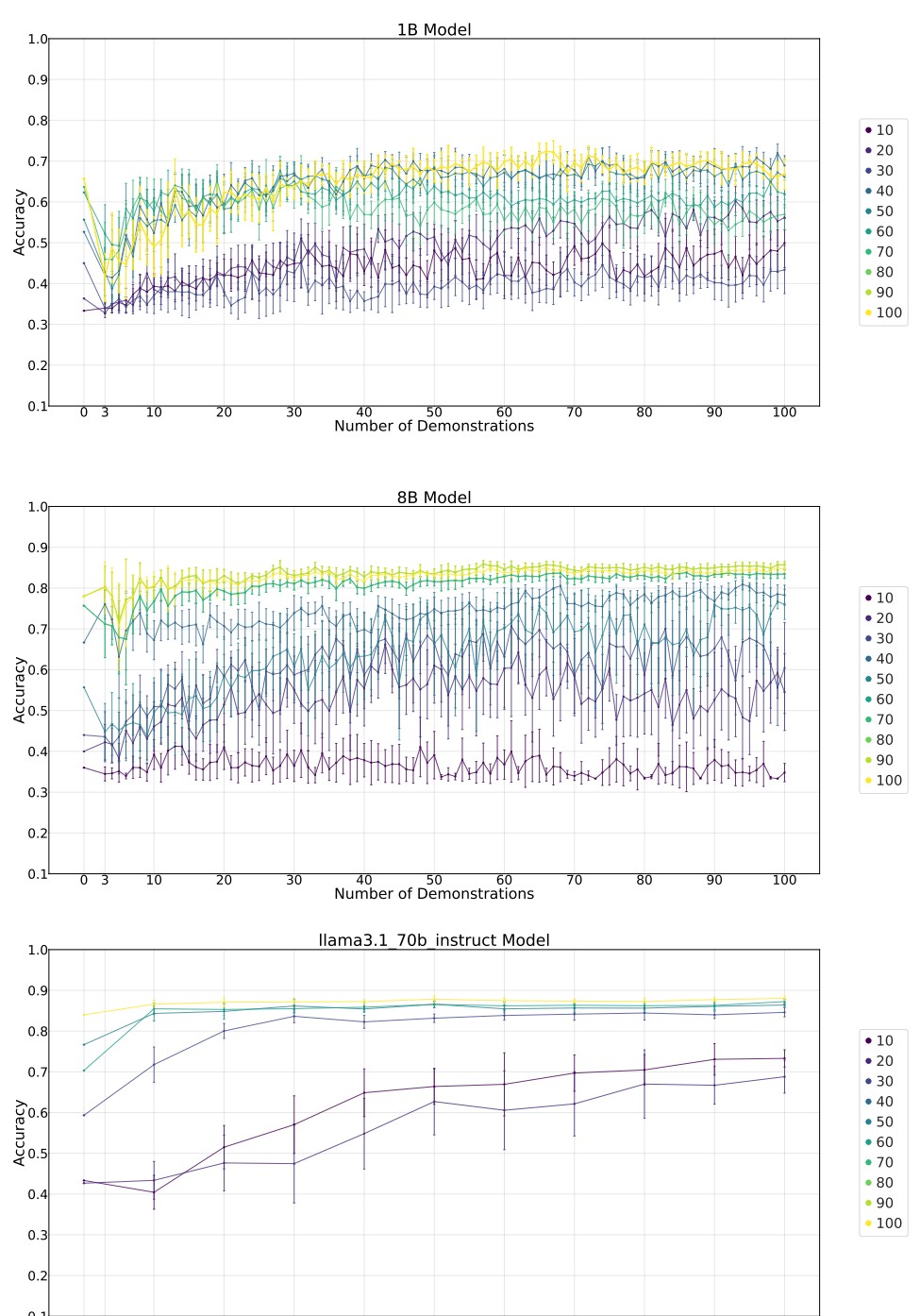

Figure 5: 3-way classification

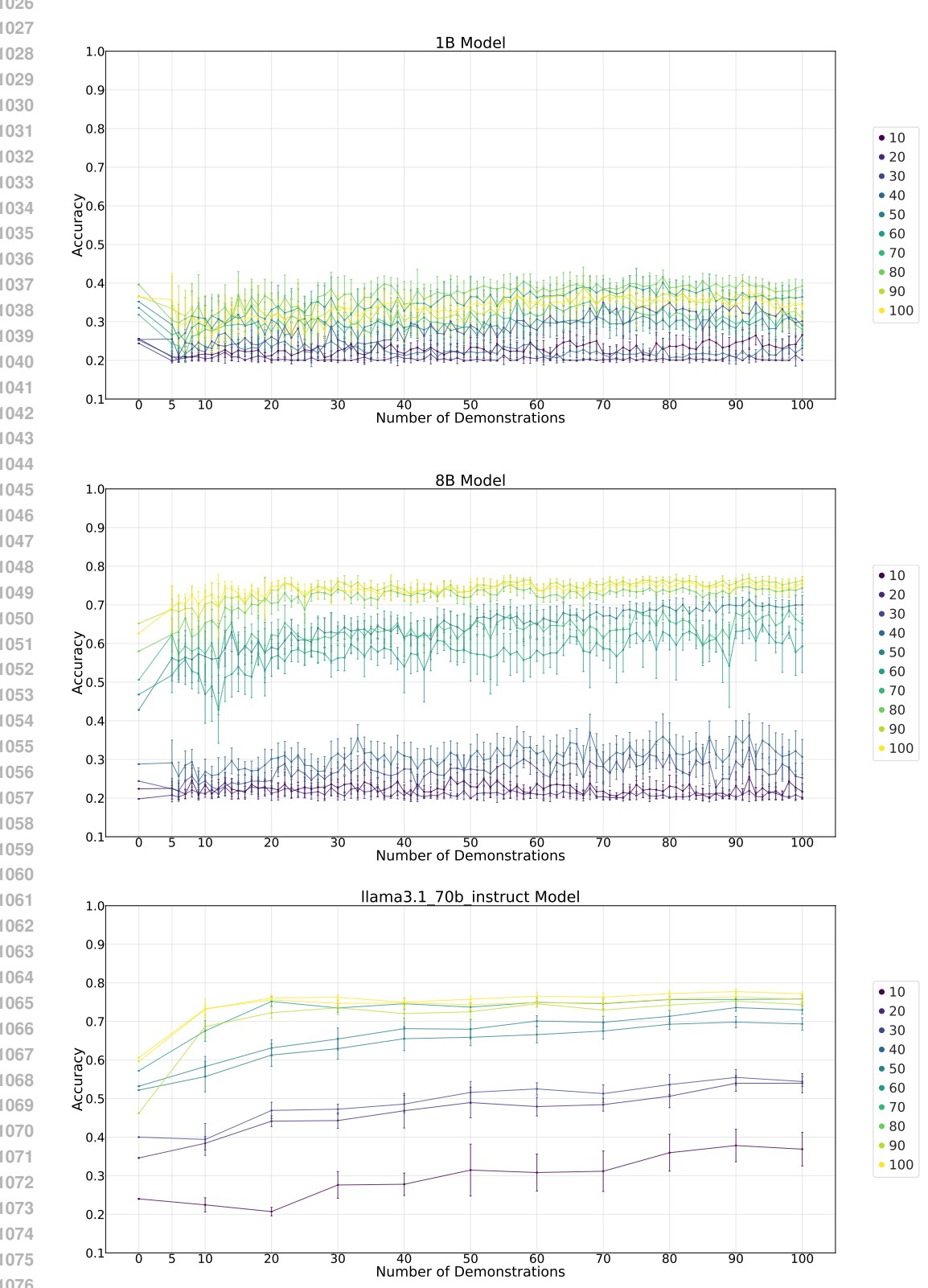

Figure 6: 5-way classification

## C ADDITIONAL MODELS: MISTRAL-7B-V0.3 AND QWEN2.5-7B

We show the label sets and learning curves for the sentiment analysis task using models from different families Mistral-7B-v0.3 and Qwen2.5-7B for 3-way and 5-way classification.

| K | Mistral | Qwen |
|---|---|---|
| 10 | diet, hack, wholes | Sick, offensive, Ath |
| 20 | nutrition, hack, excitement | JsonResult, parliamentary, applause |
| 30 | protein, complaint, excited | weighing, warn, congrat |
| 40 | fear, angry, insp | ArgumentError, bitch, Wonderful |
| 50 | fear, angry, joy | fears, hatred, celebration |
| 60 | fear, angry, joy | noir, bitch, Wonderful |
| 70 | panic, rage, smiles | fears, abusive, congrat |
| 80 | fear, anger, joy | nightmares, offender, luz |
| 90 | fear, angry, joy | terror, offending, brag |
| 100 | fear, angry, joy | terror, offending, brag |

Table 3: Label sets obtained from running Algorithm 1 on $K$ examples for 3-way classification

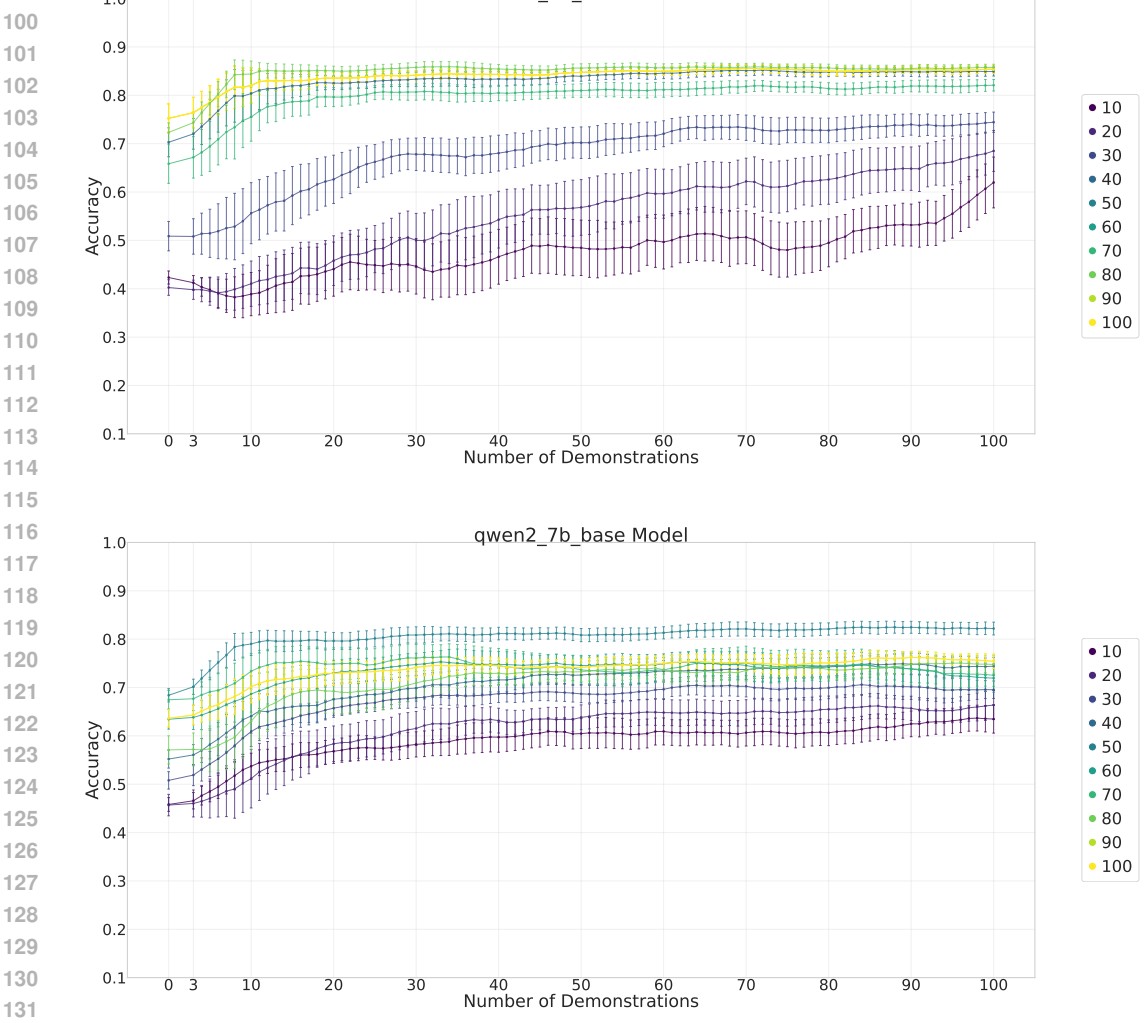

Figure 7: Mistral and Qwen 3-way classification

| K | Mistral | Qwen |
|---|---|---|
| 10 | photography, festival, AU, roll, Bull | Dialogue, SUM, clinical, Roller, negatives |
| 20 | aston, rehe, diseases, stomach, managers | Ey, academy, diagnosed, kaufen, threats |
| 30 | amazing, dress, defeat, monster, hockey | surprising, moda, diagnoses, fart, unlawful |
| 40 | surprise, competition, loss, monster, soccer | surprising, RoundedRectangle, privacy, immature, mud |
| 50 | surprise, dress, grief, monster, angry | surprising, RoundedRectangle, failures, Moo, FUCK |
| 60 | amazing, invent, depress, fright, piss | exploding, ExecutionContext, Ø§ÙHØ·, onOptionsItemSelected, misogyn |
| 70 | amazing, next, depress, terror, angry, | astonishing, Validates, distressed, nightmares, mud |
| 80 | aston, inspire, despair, fears, rage | astonishing, remains, distressed, feared, abuses |
| 90 | aston, pose, unhappy, afraid, complaint | surprising, remainder, failures, feared, bitter |
| 100 | aston, inspire, despair, fears, rage | surprising, remainder, failures, feared, bitter |

Table 4: Label sets obtained from running Algorithm 1 on $K$ examples for 5-way classification

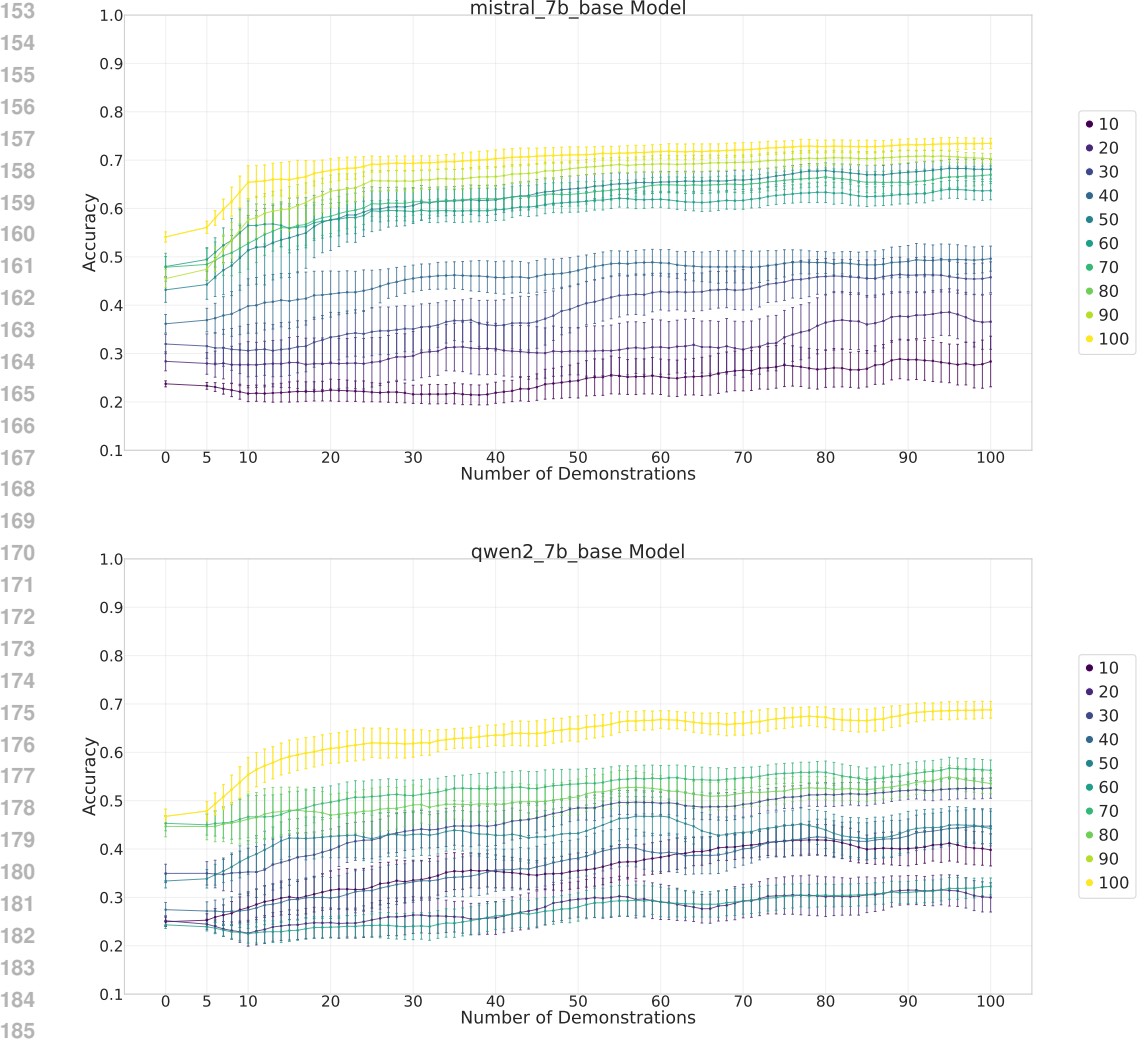

Figure 8: Mistral and Qwen 5-way classification

# D  ADDITIONAL DATASET: TREC

We conducted additional experiments using the TREC dataset (Li & Roth, 2002; Hovy et al., 2001) for question classification. The gold labels are "Entity, Description, Human, Location, Numeric." The input sentences in this dataset are questions. This makes applying our framework to this task challenging since high probability next tokens following a question would be answers rather than class names. Despite this fact, our findings from sentiment analysis hold true on the TREC dataset.

## D.1  LLAMA 3.1 8B

| K | Llama 8B |
|---|---|
| 10 | gods, Derm, Philippe, Helsinki, Wade |
| 20 | Judaism, Skin, Christians, Ukraine, Watts |
| 30 | easiest, Carb, Isaiah, Antarctica, dollars |
| 40 | GENERIC, SCC, quienes, Antarctica, dollars |
| 50 | which, Explain, who, Nations, Rate |
| 60 | Taste, explain, Malcolm, maps, Timing |
| 70 | Taste, explain, qui, maps, Timing |
| 80 | taste, development, personalities, whereabouts, ÐºÐ¾Ð»Ð¸ÑÐµÑ‚ÑÑ‚Ð²Ð¾ |
| 90 | taste, development, quienes, destinations, timed |
| 100 | Taste, explanations, qui, locations, amount |

Table 5: Label sets obtained from running Algorithm 1 on $K$ examples for 5-way classification

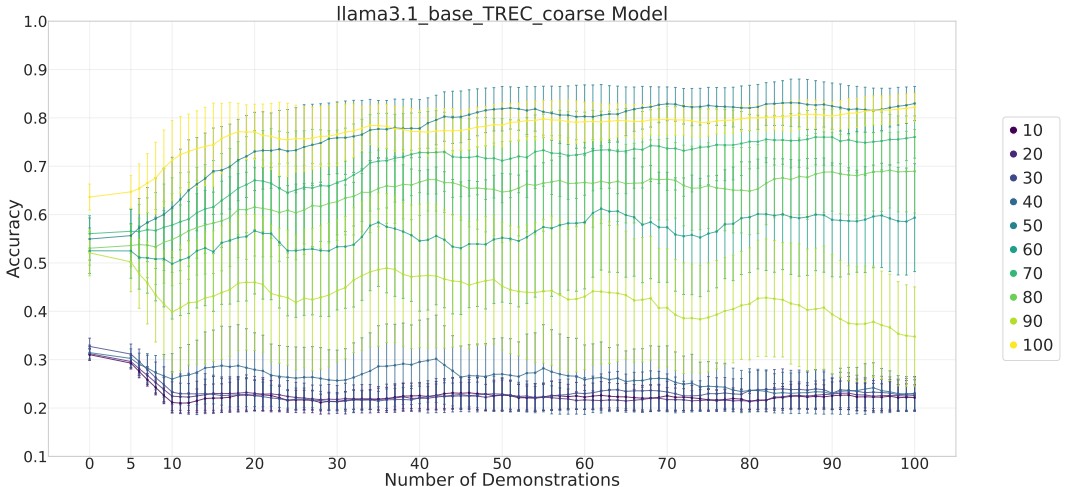

## D.2   MISTRAL 7B

| K | Mistral 7B |
|---|---|
| 10 | universe, Fer, artists, Tokyo, gover |
| 20 | Jung, advice, Leonard, Finland, college |
| 30 | matching, explaining, kings, continent, accounting |
| 40 | filling, explaining, athletes, UEFA, amount |
| 50 | night, explan, whom, UEFA, amount |
| 60 | conj, explan, whom, locations, amount |
| 70 | eating, explan, whom, locations, amount |
| 80 | drinking, explan, whom, locations, amount |
| 90 | eating, explan, whom, locations, amount |
| 100 | cul, explan, whom, locations, amount |

Table 6: Label sets obtained from running Algorithm 1 on $K$ examples for 5-way classification

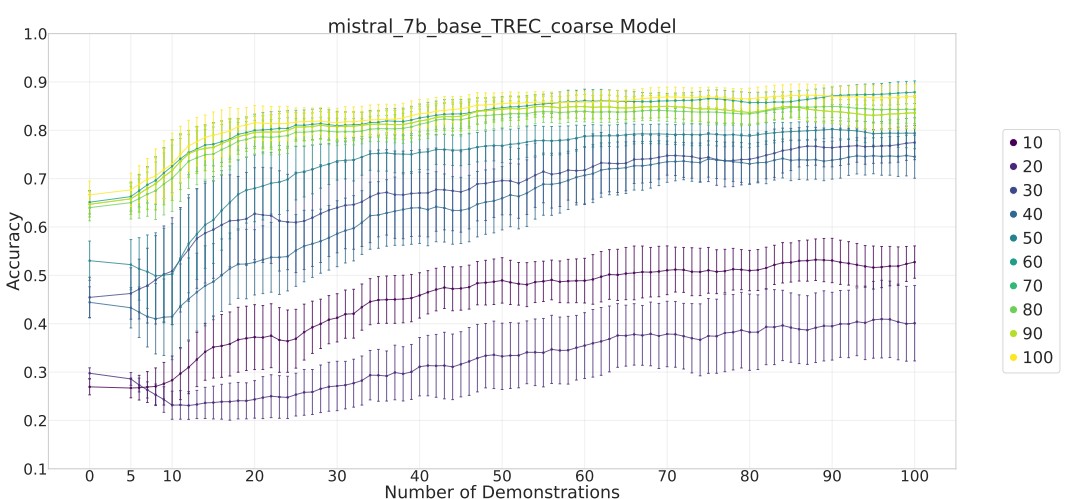

### D.3    QWEN 7B

| K | Qwen 7B |
|---|---|
| 10 | —-, Bronx, Kaz, PhoneNumber, PRES |
| 20 | comparator, Chung, Boris, wherever, æk° |
| 30 | ConfigurationManager, Geh, quoted, wherever, amounts |
| 40 | meinen, NDEBUG, Santa, wherever, amounts |
| 50 | me, Technologies, Vi, geographical, Num |
| 60 | animal, Anatomy, Vi, geographical, Num |
| 70 | animal, Lecture, Vi, geographical, Num |
| 80 | mt, Lecture, Vi, WHERE, Num |
| 90 | serif, ×¢, entreprise, gÃ©, numb |
| 100 | serif, ×¢, entreprise, gÃ©, numb |

Table 7: Label sets obtained from running Algorithm 1 on $K$ examples for 5-way classification

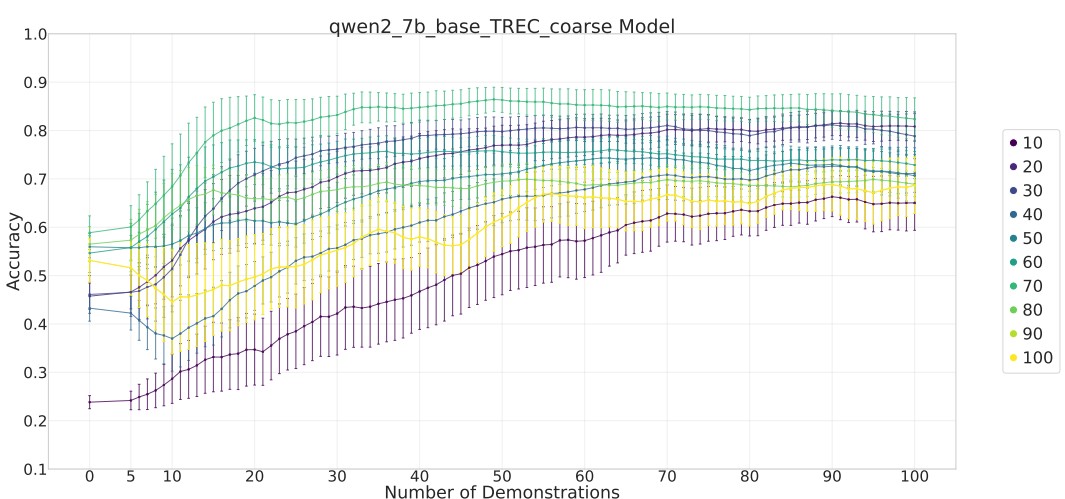

# E  IMPLEMENTATION DETAILS

## E.1  ALGORITHM 1

We first precompute the next token logits for all inputs with only one forward pass through the relabeling sentences in the dataset. During optimization, for each $\tau$ we perform a simple lookup for the logits of the $K$ sentences corresponding to the tokens in $\tau$ and normalize. The algorithm finds "good enough" solutions in the sense that it allows us to enumerate labels sets covering a wide spectrum zero-shot accuracies. Moreover, the label sets found with large $K$ are close (or identical) in meaning with the gold labels. These label sets already have a zero-shot accuracy close (within 3-5%) of the ceiling accuracy obtained after seeing demonstrations, further confirming their quality. While our algorithm is not guaranteed to find the global optimum, it is extremely simple and efficient, making it suitable for studying ICL from demonstrations or for applications which require high-quality labels.

## E.2  IN-CONTEXT LEARNING

For ICL, we sample 10 different sets of $N$ demonstrations and query the entire test set. We use prefix KV caching for demonstrations, so that during testing we only need to compute the attention between the query and the demonstrations.

**Runtime.**  We report approximate wall clock times for any fixed label set with $N$ demonstrations for 300-500 test sentences, for 10 runs on an 80GB A100 or H100 GPU: 5-7 minutes for a 7B or 8B model and 20-24h for the 70B model.

## F   CORRELATION STATISTICS FROM FIGURE 3

| $n_{demo}$ | Mean Corr. | Std Corr. | Median Corr. | CI 2.5% | CI 97.5% |
|---|---|---|---|---|---|
| 3 | 0.5486 | 0.1828 | 0.5723 | 0.1160 | 0.8336 |
| 4 | 0.6087 | 0.1933 | 0.6308 | 0.1534 | 0.9006 |
| 5 | 0.4634 | 0.2265 | 0.4970 | -0.0436 | 0.8320 |
| 6 | 0.3776 | 0.2408 | 0.4056 | -0.1363 | 0.8012 |
| 7 | 0.5743 | 0.1666 | 0.5908 | 0.2154 | 0.8582 |
| 8 | 0.4798 | 0.2303 | 0.5000 | 0.0178 | 0.8493 |
| 9 | 0.4261 | 0.1882 | 0.4356 | 0.0307 | 0.7655 |
| 10 | 0.4111 | 0.2364 | 0.4246 | -0.1043 | 0.8185 |
| 11 | 0.3826 | 0.2127 | 0.4062 | -0.0926 | 0.7447 |
| 12 | 0.5043 | 0.2660 | 0.5280 | -0.0620 | 0.8863 |
| 13 | 0.6692 | 0.1574 | 0.6770 | 0.3323 | 0.9293 |
| 14 | 0.5197 | 0.1977 | 0.5354 | 0.1040 | 0.8530 |
| 15 | 0.5170 | 0.1762 | 0.5215 | 0.1420 | 0.8431 |
| 16 | 0.4968 | 0.1781 | 0.4985 | 0.1169 | 0.8185 |
| 17 | 0.4846 | 0.1518 | 0.4862 | 0.1531 | 0.7693 |
| 18 | 0.5639 | 0.1631 | 0.5706 | 0.2400 | 0.8739 |
| 19 | 0.4597 | 0.1964 | 0.4653 | 0.0431 | 0.8037 |
| 20 | 0.6816 | 0.1389 | 0.6893 | 0.3950 | 0.9171 |
| 21 | 0.5907 | 0.1245 | 0.5846 | 0.3620 | 0.8510 |
| 22 | 0.6074 | 0.1372 | 0.6074 | 0.3508 | 0.8800 |
| 23 | 0.5401 | 0.1593 | 0.5461 | 0.2025 | 0.8406 |
| 24 | 0.5110 | 0.1682 | 0.5215 | 0.1657 | 0.8089 |
| 25 | 0.5436 | 0.1768 | 0.5583 | 0.1533 | 0.8529 |
| 26 | 0.6441 | 0.1377 | 0.6442 | 0.3754 | 0.8896 |
| 27 | 0.5369 | 0.2018 | 0.5461 | 0.1043 | 0.8897 |
| 28 | 0.5540 | 0.1569 | 0.5539 | 0.2277 | 0.8677 |
| 29 | 0.5478 | 0.1135 | 0.5354 | 0.3642 | 0.7939 |
| 30 | 0.5293 | 0.1186 | 0.5338 | 0.3374 | 0.7694 |
| 31 | 0.6273 | 0.1472 | 0.6319 | 0.3252 | 0.8896 |
| 32 | 0.5963 | 0.1561 | 0.6031 | 0.2345 | 0.8923 |
| 33 | 0.6415 | 0.1312 | 0.6442 | 0.3865 | 0.8800 |
| 34 | 0.6192 | 0.1265 | 0.6197 | 0.4000 | 0.8677 |
| 35 | 0.7148 | 0.1223 | 0.7200 | 0.4492 | 0.9142 |
| 36 | 0.7034 | 0.1209 | 0.7178 | 0.4479 | 0.9047 |
| 37 | 0.6910 | 0.1282 | 0.7017 | 0.4320 | 0.9047 |
| 38 | 0.5380 | 0.1757 | 0.5556 | 0.0983 | 0.8308 |
| 39 | 0.5583 | 0.1215 | 0.5516 | 0.3508 | 0.8062 |
| 40 | 0.6259 | 0.1396 | 0.6339 | 0.3395 | 0.8678 |

Table 8: 3-way classification, 1B model (green curve in Figure 3a). Ranking correlations across label sets for different numbers of demonstrations $n_{demo}$ (bootstrap = 1000 samples).

| $n_\text{demo}$ | Mean Corr. | Std Corr. | Median Corr. | CI 2.5% | CI 97.5% |
|---|---|---|---|---|---|
| 3 | 0.8484 | 0.0735 | 0.8528 | 0.6892 | 0.9663 |
| 4 | 0.8480 | 0.0822 | 0.8568 | 0.6647 | 0.9725 |
| 5 | 0.7627 | 0.1246 | 0.7778 | 0.4628 | 0.9540 |
| 6 | 0.7723 | 0.1528 | 0.7898 | 0.4204 | 0.9816 |
| 7 | 0.8324 | 0.0875 | 0.8396 | 0.6439 | 0.9724 |
| 8 | 0.8713 | 0.0760 | 0.8841 | 0.6944 | 0.9847 |
| 9 | 0.8937 | 0.0536 | 0.9013 | 0.7655 | 0.9754 |
| 10 | 0.8945 | 0.0599 | 0.9013 | 0.7509 | 0.9785 |
| 11 | 0.8859 | 0.0658 | 0.8924 | 0.7324 | 0.9847 |
| 12 | 0.8623 | 0.0827 | 0.8797 | 0.6647 | 0.9754 |
| 13 | 0.8540 | 0.0804 | 0.8616 | 0.6563 | 0.9754 |
| 14 | 0.8846 | 0.0577 | 0.8890 | 0.7570 | 0.9754 |
| 15 | 0.9063 | 0.0548 | 0.9136 | 0.7809 | 0.9847 |
| 16 | 0.8931 | 0.0566 | 0.8986 | 0.7654 | 0.9847 |
| 17 | 0.9045 | 0.0522 | 0.9109 | 0.7878 | 0.9816 |
| 18 | 0.9061 | 0.0605 | 0.9164 | 0.7509 | 0.9847 |
| 19 | 0.8844 | 0.0647 | 0.8948 | 0.7385 | 0.9847 |
| 20 | 0.9155 | 0.0514 | 0.9232 | 0.7902 | 0.9847 |
| 21 | 0.9117 | 0.0469 | 0.9170 | 0.8062 | 0.9816 |
| 22 | 0.8878 | 0.0644 | 0.8924 | 0.7447 | 0.9847 |
| 23 | 0.8936 | 0.0602 | 0.9013 | 0.7509 | 0.9754 |
| 24 | 0.9147 | 0.0472 | 0.9229 | 0.8037 | 0.9847 |
| 25 | 0.9026 | 0.0546 | 0.9109 | 0.7895 | 0.9816 |
| 26 | 0.9153 | 0.0455 | 0.9226 | 0.8117 | 0.9847 |
| 27 | 0.9582 | 0.0245 | 0.9630 | 0.8986 | 0.9877 |
| 28 | 0.9381 | 0.0381 | 0.9478 | 0.8493 | 0.9877 |
| 29 | 0.9282 | 0.0482 | 0.9398 | 0.8124 | 0.9877 |
| 30 | 0.9086 | 0.0521 | 0.9136 | 0.8001 | 0.9877 |
| 31 | 0.8965 | 0.0547 | 0.9011 | 0.7778 | 0.9847 |
| 32 | 0.9002 | 0.0527 | 0.9072 | 0.7901 | 0.9847 |
| 33 | 0.9498 | 0.0286 | 0.9507 | 0.8863 | 0.9877 |
| 34 | 0.9187 | 0.0491 | 0.9232 | 0.8068 | 0.9847 |
| 35 | 0.9206 | 0.0471 | 0.9260 | 0.8185 | 0.9847 |
| 36 | 0.9046 | 0.0476 | 0.9109 | 0.8025 | 0.9754 |
| 37 | 0.8970 | 0.0520 | 0.9013 | 0.7809 | 0.9754 |
| 38 | 0.9239 | 0.0445 | 0.9291 | 0.8250 | 0.9877 |
| 39 | 0.9192 | 0.0440 | 0.9259 | 0.8209 | 0.9847 |
| 40 | 0.9086 | 0.0513 | 0.9136 | 0.7878 | 0.9847 |

Table 9: 3-way classification, 8B model (purple curve in Figure 3a). Ranking correlations across label sets for different numbers of demonstrations $n_\text{demo}$ (bootstrap = 1000 samples).

| $n_\text{demo}$ | Mean Corr. | Std Corr. | Median Corr. | CI 2.5% | CI 97.5% |
|---|---|---|---|---|---|
| 10 | 0.8732 | 0.0769 | 0.8857 | 0.7143 | 1.0000 |
| 20 | 0.8978 | 0.0808 | 0.9429 | 0.7143 | 1.0000 |
| 30 | 0.8771 | 0.1039 | 0.8986 | 0.5798 | 1.0000 |
| 40 | 0.9108 | 0.0972 | 0.9429 | 0.6571 | 1.0000 |

Table 10: 3-way classification, 70B model (orange curve in Figure 3a). Ranking correlations across label sets for different numbers of demonstrations $n_\text{demo}$ (bootstrap = 1000 samples).

| $n_{\text{demo}}$ | Mean Corr. | Std Corr. | Median Corr. | CI 2.5% | CI 97.5% |
|---|---|---|---|---|---|
| 5 | 0.6375 | 0.1627 | 0.6444 | 0.2721 | 0.8910 |
| 6 | 0.6621 | 0.2044 | 0.7016 | 0.1567 | 0.9387 |
| 7 | 0.5193 | 0.2427 | 0.5636 | -0.0324 | 0.8571 |
| 8 | 0.4579 | 0.2972 | 0.4817 | -0.1626 | 0.9030 |
| 9 | 0.4426 | 0.2540 | 0.4788 | -0.1342 | 0.8390 |
| 10 | 0.5284 | 0.2252 | 0.5710 | 0.0485 | 0.8573 |
| 11 | 0.4324 | 0.2577 | 0.4602 | -0.1664 | 0.8303 |
| 12 | 0.3863 | 0.2584 | 0.3988 | -0.1030 | 0.8477 |
| 13 | 0.4129 | 0.2881 | 0.4479 | -0.2121 | 0.8788 |
| 14 | 0.5727 | 0.2140 | 0.5957 | 0.0915 | 0.8998 |
| 15 | 0.6749 | 0.1798 | 0.7091 | 0.2118 | 0.9180 |
| 16 | 0.5898 | 0.2114 | 0.6140 | 0.1090 | 0.9152 |
| 17 | 0.7062 | 0.1538 | 0.7333 | 0.3281 | 0.9362 |
| 18 | 0.5889 | 0.2328 | 0.6371 | 0.0182 | 0.9119 |
| 19 | 0.6046 | 0.1892 | 0.6322 | 0.1758 | 0.8875 |
| 20 | 0.6725 | 0.1878 | 0.7052 | 0.2438 | 0.9268 |
| 21 | 0.6258 | 0.1622 | 0.6575 | 0.2673 | 0.8754 |
| 22 | 0.5416 | 0.2144 | 0.5593 | 0.0793 | 0.8875 |
| 23 | 0.5537 | 0.2003 | 0.5888 | 0.0910 | 0.8633 |
| 24 | 0.6124 | 0.1902 | 0.6242 | 0.1877 | 0.9030 |
| 25 | 0.5332 | 0.2402 | 0.5394 | -0.0064 | 0.9030 |
| 26 | 0.5958 | 0.2098 | 0.6353 | 0.1155 | 0.9067 |
| 27 | 0.6985 | 0.1357 | 0.7091 | 0.3951 | 0.9119 |
| 28 | 0.6916 | 0.1213 | 0.7052 | 0.4423 | 0.9030 |
| 29 | 0.6130 | 0.1596 | 0.6252 | 0.2605 | 0.8754 |
| 30 | 0.7262 | 0.1624 | 0.7660 | 0.3343 | 0.9329 |
| 31 | 0.7641 | 0.1523 | 0.7939 | 0.3888 | 0.9606 |
| 32 | 0.6992 | 0.1691 | 0.7333 | 0.3100 | 0.9394 |
| 33 | 0.6159 | 0.1857 | 0.6444 | 0.1581 | 0.8997 |
| 34 | 0.6727 | 0.1698 | 0.7052 | 0.2917 | 0.9119 |
| 35 | 0.7016 | 0.1831 | 0.7576 | 0.3251 | 0.9483 |
| 36 | 0.7466 | 0.1361 | 0.7697 | 0.4133 | 0.9391 |
| 37 | 0.7843 | 0.1305 | 0.8146 | 0.4479 | 0.9545 |
| 38 | 0.7434 | 0.1212 | 0.7538 | 0.4862 | 0.9484 |
| 39 | 0.7288 | 0.1418 | 0.7516 | 0.4109 | 0.9394 |
| 40 | 0.6682 | 0.1699 | 0.6930 | 0.2606 | 0.9119 |

Table 11: 5-way classification, 1B model (green curve in Figure 3b). Ranking correlations across label sets for different numbers of demonstrations $n_{\text{demo}}$ (bootstrap = 1000 samples).

| $n_{\text{demo}}$ | Mean Corr. | Std Corr. | Median Corr. | CI 2.5% | CI 97.5% |
|---|---|---|---|---|---|
| 5 | 0.8698 | 0.0758 | 0.8788 | 0.6969 | 0.9758 |
| 6 | 0.9022 | 0.0546 | 0.9119 | 0.7669 | 0.9848 |
| 7 | 0.9227 | 0.0437 | 0.9273 | 0.8061 | 0.9879 |
| 8 | 0.9013 | 0.0593 | 0.9152 | 0.7576 | 0.9879 |
| 9 | 0.8687 | 0.0740 | 0.8815 | 0.6969 | 0.9758 |
| 10 | 0.9027 | 0.0550 | 0.9152 | 0.7573 | 0.9758 |
| 11 | 0.8831 | 0.0754 | 0.9030 | 0.7054 | 0.9849 |
| 12 | 0.8612 | 0.0773 | 0.8754 | 0.6809 | 0.9755 |
| 13 | 0.9225 | 0.0406 | 0.9273 | 0.8303 | 0.9879 |
| 14 | 0.9112 | 0.0581 | 0.9273 | 0.7697 | 0.9879 |
| 15 | 0.9331 | 0.0353 | 0.9394 | 0.8545 | 0.9879 |
| 16 | 0.9222 | 0.0419 | 0.9273 | 0.8207 | 0.9879 |
| 17 | 0.9276 | 0.0353 | 0.9362 | 0.8509 | 0.9879 |
| 18 | 0.9045 | 0.0469 | 0.9152 | 0.7939 | 0.9758 |
| 19 | 0.8882 | 0.0604 | 0.9030 | 0.7333 | 0.9758 |
| 20 | 0.9530 | 0.0275 | 0.9515 | 0.8908 | 0.9970 |
| 21 | 0.9438 | 0.0370 | 0.9515 | 0.8510 | 0.9970 |
| 22 | 0.9449 | 0.0306 | 0.9515 | 0.8788 | 0.9879 |
| 23 | 0.9458 | 0.0308 | 0.9515 | 0.8788 | 0.9879 |
| 24 | 0.9453 | 0.0263 | 0.9483 | 0.8875 | 0.9879 |
| 25 | 0.9227 | 0.0354 | 0.9273 | 0.8449 | 0.9758 |
| 26 | 0.9374 | 0.0345 | 0.9394 | 0.8667 | 0.9879 |
| 27 | 0.9236 | 0.0393 | 0.9273 | 0.8303 | 0.9879 |
| 28 | 0.9289 | 0.0378 | 0.9362 | 0.8424 | 0.9879 |
| 29 | 0.9128 | 0.0468 | 0.9165 | 0.8060 | 0.9818 |
| 30 | 0.9524 | 0.0249 | 0.9515 | 0.8909 | 0.9879 |
| 31 | 0.9411 | 0.0311 | 0.9423 | 0.8667 | 0.9879 |
| 32 | 0.9385 | 0.0332 | 0.9394 | 0.8667 | 0.9879 |
| 33 | 0.9466 | 0.0257 | 0.9515 | 0.8909 | 0.9879 |
| 34 | 0.9403 | 0.0297 | 0.9394 | 0.8788 | 0.9879 |
| 35 | 0.9480 | 0.0295 | 0.9515 | 0.8788 | 0.9879 |
| 36 | 0.9223 | 0.0392 | 0.9273 | 0.8424 | 0.9879 |
| 37 | 0.9429 | 0.0281 | 0.9483 | 0.8788 | 0.9879 |
| 38 | 0.9320 | 0.0414 | 0.9394 | 0.8292 | 0.9879 |
| 39 | 0.9443 | 0.0306 | 0.9483 | 0.8788 | 0.9879 |
| 40 | 0.9371 | 0.0315 | 0.9394 | 0.8667 | 0.9940 |

Table 12: 5-way classification, 8B model (purple curve in Figure 3b). Ranking correlations across label sets for different numbers of demonstrations $n_{\text{demo}}$ (bootstrap = 1000 samples).

| $n_{\text{demo}}$ | Mean Corr. | Std Corr. | Median Corr. | CI 2.5% | CI 97.5% |
|---|---|---|---|---|---|
| 10 | 0.8701 | 0.0744 | 0.8833 | 0.7000 | 0.9500 |
| 20 | 0.8955 | 0.0476 | 0.9000 | 0.7500 | 0.9500 |
| 30 | 0.8549 | 0.0776 | 0.8833 | 0.7000 | 0.9542 |
| 40 | 0.8683 | 0.0730 | 0.8833 | 0.6946 | 0.9667 |

Table 13: 5-way classification, 8B model (orange curve in Figure 3b). Ranking correlations across label sets for different numbers of demonstrations $n_{\text{demo}}$ (bootstrap = 1000 samples).

# G  CORRELATION STATISTICS FROM FIGURE 4

| $K$ | Mean Corr. | Std Corr. | Median Corr. | CI 2.5% | CI 97.5% |
|---|---|---|---|---|---|
| **1B** | | | | | |
| 10 | 0.5931 | 0.0986 | 0.6029 | 0.3822 | 0.7745 |
| 20 | 0.5579 | 0.0975 | 0.5615 | 0.3536 | 0.7347 |
| 30 | 0.2003 | 0.1232 | 0.2018 | -0.0451 | 0.4343 |
| 40 | 0.6339 | 0.0713 | 0.6428 | 0.4859 | 0.7570 |
| 50 | 0.6622 | 0.0765 | 0.6675 | 0.4997 | 0.7909 |
| 60 | 0.3582 | 0.1165 | 0.3626 | 0.1221 | 0.5709 |
| 70 | 0.2686 | 0.1195 | 0.2741 | 0.0325 | 0.4863 |
| 80 | 0.5777 | 0.0847 | 0.5819 | 0.4086 | 0.7279 |
| 90 | 0.5821 | 0.0864 | 0.5867 | 0.3983 | 0.7435 |
| 100 | 0.5818 | 0.0824 | 0.5873 | 0.4023 | 0.7308 |
| **8B** | | | | | |
| 10 | 0.0636 | 0.1384 | 0.0677 | -0.2137 | 0.3244 |
| 20 | 0.4146 | 0.1236 | 0.4192 | 0.1486 | 0.6373 |
| 30 | 0.5769 | 0.1038 | 0.5840 | 0.3687 | 0.7708 |
| 40 | 0.1968 | 0.1308 | 0.1924 | -0.0673 | 0.4537 |
| 50 | 0.5073 | 0.1236 | 0.5169 | 0.2436 | 0.7229 |
| 60 | 0.5840 | 0.0941 | 0.5892 | 0.3956 | 0.7597 |
| 70 | 0.5785 | 0.1011 | 0.5875 | 0.3644 | 0.7570 |
| 80 | 0.4405 | 0.1200 | 0.4424 | 0.1980 | 0.6622 |
| 90 | 0.4433 | 0.1153 | 0.4448 | 0.2089 | 0.6690 |
| 100 | 0.3338 | 0.1289 | 0.3373 | 0.0912 | 0.5783 |
| **70B** | | | | | |
| 10 | 0.7446 | 0.2272 | 0.8000 | 0.2052 | 1.0000 |
| 20 | 0.3503 | 0.4338 | 0.3000 | -0.6000 | 0.9000 |
| 30 | 0.8398 | 0.1380 | 0.9000 | 0.4000 | 1.0000 |
| 40 | 0.6134 | 0.2642 | 0.7000 | 0.0513 | 1.0000 |
| 60 | 0.5237 | 0.2925 | 0.6000 | 0.0000 | 1.0000 |
| 70 | 0.5931 | 0.2530 | 0.6156 | 0.1000 | 1.0000 |

Table 14: 3-way classification correlations (between N and $N$-shot accuracy) from Figure 4a for 1B (green curve), 8B (purple curve), and 70B (orange curve) models across different $K$ values (bootstrap = 1000 samples). Each $K$ value corresponds to a learning curve, which is determined by its zero-shot accuracy in the figure.

| $K$ | Mean Corr. | Std Corr. | Median Corr. | CI 2.5% | CI 97.5% |
|---|---|---|---|---|---|
| | | | **1B** | | |
| 10 | 0.1794 | 0.1300 | 0.1774 | -0.0803 | 0.4313 |
| 20 | -0.2500 | 0.1429 | -0.2555 | -0.5172 | 0.0297 |
| 30 | 0.1871 | 0.1247 | 0.1825 | -0.0409 | 0.4303 |
| 40 | 0.0607 | 0.1257 | 0.0557 | -0.1881 | 0.3077 |
| 50 | 0.1119 | 0.1278 | 0.1156 | -0.1320 | 0.3608 |
| 60 | 0.3877 | 0.1190 | 0.3982 | 0.1376 | 0.5956 |
| 70 | 0.2916 | 0.1202 | 0.2941 | 0.0488 | 0.5246 |
| 80 | 0.3405 | 0.1178 | 0.3437 | 0.1047 | 0.5689 |
| 90 | 0.1268 | 0.1308 | 0.1282 | -0.1364 | 0.3775 |
| 100 | 0.2378 | 0.1352 | 0.2406 | -0.0256 | 0.4862 |
| | | | **8B** | | |
| 10 | 0.0798 | 0.1523 | 0.0830 | -0.2172 | 0.3850 |
| 20 | 0.0953 | 0.1268 | 0.0940 | -0.1428 | 0.3399 |
| 30 | 0.4461 | 0.1226 | 0.4552 | 0.1838 | 0.6608 |
| 40 | 0.3511 | 0.1377 | 0.3550 | 0.0710 | 0.6043 |
| 50 | 0.5517 | 0.1062 | 0.5585 | 0.3444 | 0.7421 |
| 60 | 0.5076 | 0.1090 | 0.5127 | 0.2809 | 0.7131 |
| 70 | 0.4112 | 0.1039 | 0.4144 | 0.1865 | 0.5979 |
| 80 | 0.6527 | 0.0810 | 0.6579 | 0.4772 | 0.7990 |
| 90 | 0.5821 | 0.0931 | 0.5829 | 0.3833 | 0.7503 |
| 100 | 0.4269 | 0.1062 | 0.4301 | 0.2116 | 0.6296 |
| | | | **70B** | | |
| 10 | 0.3457 | 0.3613 | 0.4000 | -0.5643 | 0.9000 |
| 20 | 0.7973 | 0.1862 | 0.9000 | 0.3000 | 1.0000 |
| 30 | 0.7250 | 0.1941 | 0.8000 | 0.2051 | 1.0000 |
| 40 | 0.7799 | 0.1960 | 0.8000 | 0.2000 | 1.0000 |
| 50 | 0.8705 | 0.1301 | 0.9000 | 0.6000 | 1.0000 |
| 60 | 0.7653 | 0.1697 | 0.7000 | 0.3000 | 1.0000 |
| 70 | 0.6708 | 0.2027 | 0.7000 | 0.3000 | 1.0000 |
| 90 | 0.6664 | 0.2613 | 0.7000 | 0.1000 | 1.0000 |
| 100 | 0.5465 | 0.2877 | 0.6000 | 0.0513 | 1.0000 |

Table 15: 5-way classification correlations from Figure 4b for 1B (green curve), 8B (purple curve), and 70B (orange curve) models across different $K$ values (bootstrap = 1000 samples). Each $K$ value corresponds to a learning curve, which is determined by its zero-shot accuracy in the figure.

