# OpenReview forum: "On the Relationship Between the Choice of Representation and In-Context Learning"
_ICLR.cc/2026/Conference — Submitted to ICLR 2026_

### Official Review · Reviewer_sH7g · 2025-10-21

**Soundness:** 3
**Presentation:** 3
**Contribution:** 3
**Rating:** 6
**Confidence:** 4

**Summary:**

This paper explores the interplay between label representation and in-context learning (ICL) in LLMs, positing their independence: representation establishes baseline accuracy, while learning builds upon it. It introduces an optimization algorithm to generate label sets of varying semantic relevance, tested on sentiment classification with Llama models (1B, 8B, 70B parameters). Results confirm consistent label rankings, representation-constrained accuracy, and learning influenced by model size and label quality. It's a valuable addition to ICL research with innovative methods, but restricted to classification and synthetic data.

**Strengths:**

1. Rigorous hypothesis formulation. The independence of representation (zero-shot baseline) and demonstration learning resolves prior ICL inconsistencies (e.g., Pan et al., 2023; Kirsanov et al., 2025), offering a logical framework for prompt decomposition.

2. Innovative methodological approach. Hill-climbing optimization enumerates label sets by semantic relevance, surpassing binary gold/abstract schemes for precise representation quantification in classification.

3. Comprehensive empirical analysis. Varies demonstrations (up to 100), model scales, and tasks (3/5-way), using slopes, correlations, and bootstrapping to prove orthogonality and reveal model/label conditioning on efficiency.

**Weaknesses:**

1. Limited dataset diversity. Reliance on a single synthetic sentiment dataset (1,000 sentences) may not capture real-world variability, potentially overstating the generalizability of findings.

2. Optimization algorithm simplicity. The hill-climbing approach for label enumeration, while effective, lacks comparison to more advanced methods (e.g., genetic algorithms) and may miss global optima.

3. Incomplete resource reporting. No details on computational costs, such as training epochs, hardware requirements, or runtime, hindering reproducibility for resource-constrained researchers.

4. Narrow task scope. Experiments focus solely on 3-way and 5-way classification; no exploration of multi-label, regression, or non-text tasks to assess broader ICL implications.

5. Underexplored edge cases. While overlap in label semantics is noted, the paper does not address or mitigate potential biases from prompt formatting variations or noisy demonstrations.

**Questions:**

See Weaknesses.

---

> ### Author Response · Authors · 2025-11-17
> **Rebuttal**
>
> Thank you for the review and questions! Below, we will discuss each point:
> >Limited dataset diversity
>
> We chose to focus on one dataset so that we could study many experimental conditions: different model sizes, varying the number of demonstrations up to 100, and 3-way and 5-way classification. Sentiment analysis is a commonly used task in NLP and many previous studies used it (Pan et al. 2023, Min et al. 2022). Moreover, it was convenient that we could study both 3-way and 5-way classification on the same dataset, so we could quantify how the number of classes influences the task and findings (5-way is indeed the harder task). We are ran an experiment on a different dataset to address this point and show the results in Appendix D of the updated pdf.
>
> >Optimization algorithm simplicity
>
> The algorithm is used as an enumeration tool - it simply gives out a list of label sets, ranging from random tokens as labels (which were obtained when we ran with small K), to labels that are comparable to gold labels (obtained for large K). We do not need the global optima. As shown in Figure 2, the label sets obtained with K=100 have a high enough zero-shot accuracy. We also want to emphasize its simplicity and efficiency - only one forward pass for each input in the relabeling sentences set, which makes it practical for any kind of compute resources, as opposed to more costly evolutionary algorithms.
>
> >Incomplete resource reporting
>
> We will add a section in the appendix discussing the compute resources. Since we are not training any models (we are only doing inference) and our code will be made publicly available, we did not have any concerns about reproducibility, but will discuss it in the paper for more transparency.
>
> >Narrow task scope
>
> We acknowledge that the focus on one token label is a limitation and we plan to add a section discussing it and suggesting future work. Moreover, we are focused on classification, and other tasks such as regression would involve a different research question. The output of regression is a number, so there is only one way to represent that. We are also focusing on language and language-only models specifically, so non-text data would be left for future work, as it is already represented in a different modality and it is unclear what variations to the representation to use.
>
> >Underexplored edge cases
>
> We chose a minimal prompt formatting (as shown in Figure 1) to minimize any variations other than those from the label space. We considered the variation from inputs and for each N number of demonstrations, we randomly sampled 10 sets of demonstrations from the demonstration dataset. The error bars in Figure 2,3, and 4 reflect that.
>
> Please let us know if our response addresses your concerns and if we have understood your questions correctly. Thank you!

---

### Official Review · Reviewer_cjXy · 2025-10-30

**Soundness:** 3
**Presentation:** 3
**Contribution:** 2
**Rating:** 4
**Confidence:** 3

**Summary:**

This paper investigates the relationship between representation and learning in ICL tasks. Through extensive experiments and detailed analyses, the authors demonstrate that representation and learning are largely independent (orthogonal) to each other.

**Strengths:**

1.The study covers LLMs of various sizes, from 1B to 70B parameters, which validates the scalability of the findings.

2.The analysis is thorough and deep, incorporating correlation studies and discussing the effect of zero-shot performance on final results.

3.Most methodological details and experimental settings are clearly described.

**Weaknesses:**

My primary concern lies in the generalizability of the findings. The scope of experiments and the refined label settings raise questions about how broadly the conclusions can be applied. Specifically:

1.Since the experiments are conducted solely on sentiment analysis, it remains unclear whether the findings extend to other ICL tasks.

2.The label set refinement process (Algorithm 1) may limit generalizability. Because the label names were carefully selected and are not the original class names, the conclusions might only hold under these controlled conditions. In real-world ICL scenarios, where original class names are typically used, will the observed independence between representation and learning still persist?

3.It would also be valuable to understand the relationship between representation and learning when randomly selected class names are used instead of refined ones.

I am open to revising my evaluation if the authors can adequately address these concerns.

**Questions:**

1.What specific prompt template was used in the experiments?

2.Can the findings be applied to real-world use cases or algorithms, or are they mainly of theoretical interest?

---

> ### Author Response · Authors · 2025-11-17
> **Rebuttal**
>
> Thank you for the review and questions! We will address each of them below.
>
> >experiments are conducted solely on sentiment analysis
>
> We chose to focus on one dataset so that we could study many experimental conditions: different model sizes, varying the number of demonstrations up to 100, and 3-way and 5-way classification. Sentiment analysis is a commonly used task in NLP and many previous studies used it (Pan et al. 2023, Min et al. 2022). Moreover, it was convenient that we could study both 3-way and 5-way classification on the same dataset, so we could quantify how the number of classes influences the task and findings (5-way is indeed the harder task). We are ran an experiment on a different dataset to address this point and show the results in Appendix D of the updated pdf.
>
> >The label set refinement process (Algorithm 1) may limit generalizability
>
> We are not sure we understand this point. Would it be possible to explain more? The algorithm is used as an enumeration tool - it simply gives out a list of label sets, ranging from random tokens as labels (which were obtained when we ran with small K), to gold labels (obtained for large K). Our method for testing the hypothesis is to evaluate how the learning curves look under a variety of different labels. We can only claim that learning and representation are disentangled due to observing all of these curves. We could have not said that just by looking at a single curve when gold labels are used.
>
> >randomly selected class names are used instead of refined ones
>
> The lowest curve is obtained by running Algorithm 1 on K=10 development sentences and effectively overfits to those, resulting in random tokens such as “biomedical, malware, cloudy.” They act as random class names and the model performs at chance when asked to classify a sentence into one of these categories without any demonstrations.
>
> >What specific prompt template was used in the experiments?
>
> The template is exactly the one shown in Figure 1.
>
> >Can the findings be applied to real-world use cases or algorithms, or are they mainly of theoretical interest?
>
> The contributions of the paper are both theoretical and practical: we improve upon the empirical understanding of ICL by showing the disentanglement between learning and representation. A deeper understanding of the topic allows us to improve performance in applications. For example, using Algorithm 1, a user can select a set of labels that will maximize performance in their classification task.
>
> Looking forward to your response and we hope these answers will help clarify the points raised! Thank you again for the review!

---

### Official Review · Reviewer_cGKP · 2025-11-01

**Soundness:** 3
**Presentation:** 2
**Contribution:** 2
**Rating:** 4
**Confidence:** 3

**Summary:**

This paper investigate how the choice of label representation influences in-context learning(ICL) in large language model(LLM), and whether representation and learning from given demonstrations are independent. The authors claim that prompt's label representation and learning from given demonstrations are independent of one another and the label representation determines the zero-shot performance while learning from given demonstrations improves the performance on top of the zero-shot performance. To validate author's claim, author propose optimization algorithm for label selection by exploiting Hill-climbing algorithm. With this algorithm, the authors control the quality of label by adjusting various number of example(K) of the labeling set as an input for label selection algorithm. The authors test their hypothesis with 3-way and 5-way sentiment classification task using Llama 3 models with different size.

While the paper presents interesting analysis about the relationship between the choice of representation and in-context learning, most of core insight are evolutionary given from prior works (Min et al, Pan et al, Kirsanov et al., McCoy et al., Chen et al.) Nonetheless, the authors provide interesting and novel claim, which the choice of representation and in-context learning are independent of one another. Still, from the result, their claim seems overgeneralization. That is, the choice of representation and in-context learning are disentangle-able but it seems bit hard to say that they are independent of one another.

**Strengths:**

1. *Novel and Important Research Questions*: The paper attempts to tackle an underexplored aspect of in-context learning(ICL) by disentangling two fundamental elements of ICL, representation and learning. Prior works had mainly examined these two elements separately, while this paper examine the relationship between these two aspects and how these two aspects influence and act in ICL.

2. *Comprehensive Experimental Design*: The authors validate their hypothesis with rigor across multiple factors. Therefore, experimental evaluation is thorough and convincing. The paper covers different model size with two different classification scenarios. Also, the paper present the results with different representation quality based on labeling set and different number of demonstration. These comprehensive experiment provide multidimensional insights.

3. *Useful Insight regarding ICL*: The paper provides useful insight regarding ICL such as ranking preservation across different number of demonstration which emphasizes the importance of label representation selection.

4. *Insightful Conceptual Connection with traditional ML*: The paper provide insightful conceptual bridge with traditional ML by analogizing the choice of label representation in ICL as feature selection in traditional ML. Similar to neural network based classifier where good features enable efficient learning, a good representation provides high baseline performance in ICL.

**Weaknesses:**

1. *Critical overgeneralization of "Independence" Claim*: The major concerns with this paper was the contradiction between the main claim of "independence/orthogonality" and the experiment result. If the learning and the choice of representation are independent of one another, then the efficiency of learning(Slope in Figure 2) should not depend on the quality of the label representation. Also, the author explicitly states that "learning is conditioned by representation" and "independent but interwined effect" which seems self-contradiction of paper's core claim. The paper has successfully shown that the choice of representation and the learning are disentangle-able while it seems to fail to show that they are independent of one another.

2. *Model Size influence Claim*: From section 5.2, the paper claims that model size influences the learning rate. While the result supports its claim, if the result of using same label representation with different models is provided, the claim would have been more convincing.  From the result, we can notice that the model size and label representation influence the learning rate but we are not able to see how much model size influence and how much label representation influence the learning rate by itself. To support the claim powerfully, the result of using same label with different model should be presented.

3. *Require more explanation with some result interpretations*: While the paper provides some useful insight regrading ICL, some of the result interpretation may require more details. In section 5.2, the paper claims that the representation with a medium zero-shot accuracy benefit the most from demonstration which seems crucial finding regarding ICL. Still, there should be more explanation with this claim such as why medium zero-shot accuracy representation benefit the most. Also, in the same section, the paper explain the phenomenon observed in 1B model. The paper briefly explain that the model was confused by Nepali world. This brief explanation seem bit insufficient and left me question that for the small model, high-prior label can interfere with ICL at small number of demonstrations(Fundamental Instability of small model). To clear the this question, there should be more detail explanation or other experiment result.

**Questions:**

Q1. See weakness discussed above

Q2. The paper present the result only using Llama 3 family. I was just wondering how the result would be present using different LLM such as GPT, Qwen, Deepseek etc.

---

> ### Author Response · Authors · 2025-11-17
> **Rebuttal**
>
> Thank you for your review and constructive feedback! Below we respond to each of the weaknesses and questions.
>
> > Critical overgeneralization of "Independence" Claim
>
> We agree with you that our paper shows the contribution representation and learning can be disentangled but they are not fully independent (due to, for example, different slopes). Our initial hypothesis was that they are independent but the results show a more nuanced relationship. We are planning to revise our introduction such that this is clear.
>
> >Model Size influence Claim
>
> The actual content (and human understanding) of a given label set does not matter for our claims and results. A label set is defined by its zero-shot accuracy, therefore, evaluating two different models with the exact same label set is not necessary. If we choose label sets of similar zero-shot accuracy for different models and observe that the learning curves differ, while they started around the same point, we can conclude that model size is the factor that influenced the slope.
>
> >Require more explanation with some result interpretations
>
> The explanations of the findings about medium and low zero-shot accuracy label sets are our hypotheses. We believe that it is possible that a medium-meaningful label set is suggestive enough that the models have some hints about what task they are doing (in this case sentiment classification) and with multiple examples they learn the correct clustering. Moreover, the explanation regarding the exception is also a hypothesis, and we will further elaborate it in the paper: in the zero-shot case, there is only one test sentence, and no labels appear in the prompt. The model simply predicts the high probability token (even if it is in a different language than the input) since it does not “know” that the other labels are in a different language . As soon as demonstrations are shown (N>=3), the all three labels appear, so one of them being in a different language than the rest adds the complexity of translation to the original classification task.
>
> >Q2. The paper present the result only using Llama 3 family.
>
> We added new experiments which further validate our hypothesis with models Mistral-7B-v0.3 and Qwen2.5-7B on both 3-way and 5-way classification, for up to 100 demonstrations. We show the labeling schemes and learning curves in Appendix C of the updated pdf.
>
> Looking forward to your response, and please let us know if our answers help clear your concerns! Thank you!

---

### Official Review · Reviewer_jKgr · 2025-11-02

**Soundness:** 3
**Presentation:** 3
**Contribution:** 4
**Rating:** 6
**Confidence:** 4

**Summary:**

The paper proposes that in-context learning performance is conditioned on both the ability to learn from demonstrations ("learning") and the naturalness of the representation of the candidate classes ("representation"). They propose an optimization approach for generating label sets by iteratively hill-climbing on the task of selecting a token to use as the label for one of the classes in a zeroshot classification setting. By performing this optimization with more or less examples, they construct better or worse label sets for the same underlying classification task, and show that (1) representations that have worse zeroshot performance are consistently worse even in 100-shot settings; (2) representations that have sufficiently poor zeroshot performance (depending on the task) inhibit learning from additional demonstrations; (3) different model sizes in the same family exhibit different learning behavior, and optimizing label sets for larger models results in more semantically meaningful representations.

**Strengths:**

S1. This is a nice hypothesis and a really clear untangling of two previously conflated components of ICL; I really like the idea of varying the representation of the label set to study learning behavior on the same data. The findings are interesting and a meaningful contribution to the empirical understanding of ICL.

S2. The paper takes care with validating claims, and makes many good choices in the experimental design-- e.g the averaging over 10 demonstration sets for each ICL data point, the confidence intervals, the design of the optimization with random restarts. ICL is very noisy, but these details give me much more confidence that these results are meaningful.

S3. The presentation is generally quite strong, with meaningful engagement with prior work. The core ideas of the paper are clearly explained; I think it would be easy to follow even if I did not work in this area, and I enjoyed reading the paper!

**Weaknesses:**

W1. Many real-world label sets include multi-token labels and longer semantically meaningful labels may be better representations than any single token would allow (and allow for distinguishing between similar classes, e.g. two labels in Banking-77 are "card payment wrong exchange rate" and "card payment not accepted"). Allowing for multi-token labels clearly expands the optimization space to an absurd degree, so I'm not asking for you to incorporate this, but I think this should be discussed as a limitation-- you are only studying the space of single-token label representations. Similarly, the focus on labels prevents consideration of representational effects from other parts of the input formatting. Discussion of these limitations would make it clearer what evidence the paper can and cannot provide for the representation vs learning hypothesis.

W2. The paper considers only one family of models; while I don't think repeating every experiment across many families with many model sizes is really worth the expense, different model families do behave differently in many ways. Performing a small sweep of models at similar size across different model families (e.g. Qwen 2.5 7B and Mistral 7B, or any other pair of roughly 8B models) would help validate that this is not a Llama-family-specific effect. This is the main reason my score for the paper is not higher; if you can demonstrate this holds across model families, I'd be happy to raise my score.

W3. While the paper presents the optimization of label space mostly as a tool to study representations, it's an interesting idea on its own and would be much more impactful if there was a bit more detail. Specifically: (1) can you show on Figure 2 how these numbers compare to the baselines of semantically meaningless (e.g. numerical) labels and gold (from the original dataset) labels? (2) Figure 2 shows sample efficiency for the optimization pipeline, but can you also discuss cost in terms of compute efficiency (e.g. in average forward passes through the model during optimization?) Wall-clock time would also be a helpful metric to build intuition, though of course this depends strongly on your inference setting.

**Questions:**

Q1. I'm really interested in the transferability of these learned label sets across models (in the same family, but especially across families). Are label sets learned by one model necessarily strong label sets for another? It seems that this might be a one-way transfer (e.g. the semantically meaningful label sets from the 70B might transfer well to the 1B, but some of the 1B sets could be reflections of training artifacts that might not be mirrored in the 70B).

Q2. I could imagine a number of alternate ways of choosing different-quality label sets, including using earlier versions of the label set from the optimization process or choosing the bottom assignment out of the 10 runs at a fixed $K$. Why construct the different-quality label sets through using different-sized $K$?

Comment, not related to score: generally I like the way the results are structured into paragraphs, but I think within some of the paragraphs, the framing of "1B did this; 7B did this; 70B did this" is sometimes a little hard to follow. Reframing this to focus on trends in results first and then discussing where each result applied would make it easier to read: "generally, either X or Y happened; X happened more for [some models over others]". I noticed this most in lines 398-403. By contrast, I liked the presentation in lines 357-364.

---

> ### Author Response · Authors · 2025-11-17
> **Rebuttal**
>
> Thank you for your positive review and useful feedback! We respond to each of the concerns raised below:
>
> >W1. Many real-world label sets include multi-token labels
>
> We acknowledge that the focus on single token label is a limitation and we will add a section discussing its implications and potential future work extensions to multi-label. In addition, we will discuss the choice of focusing on the label space as opposed to the input space and other format features of the prompt.
>
> >W2. The paper considers only one family of models
>
> We conducted new experiments which further validate our hypothesis with models Mistral-7B-v0.3 and Qwen2.5-7B on both 3-way and 5-way classification, for up to 100 demonstrations. We show the labeling schemes and learning curves in Appendix C of the updated pdf.
>
> >W3. optimization of label space
>
> It is unclear if W3 refers more to the actual algorithm 1, or to the ICL part, which is separate and comes after the labels sets have been computed. We respond to both: (1) The implementation of Algorithm 1 precomputes the logits for all inputs (only one forward pass through the relabeling sentences) and just does indexing during optimization, so it is extremely efficient. We will add a note on this in the paper (we will also release our code). Regarding the meaningless and meaningful labels, the K=10 curve is equivalent to abstract/numerical labels (they have similar zero-shot accuracy) and the optimization with K=100 typically finds the gold label, so the K=100 learning curve in Figure 2 reflects this. (2)The ICL part requires a forward pass for each label set and each number of demonstrations. We will add some example wall clock times and details about the hardware.
>
> >Q1. transferability of these learned label sets across models
>
> First, as mentioned above Algorithm 1 is extremely efficient and requires K=number of relabeling sentences forward passes so it is feasible to do it for any desired model. Second, our hypothesis is that ICL performance depends on $p_{\theta}(token|input)$, which depends primarily on the pretraining data of each model $\theta$. Since we cannot access the pretraining data, we cannot claim that these probabilities should be similar (even across the same model family). Furthermore, one can simply verify with their current model the zero-shot accuracy of any label set (no matter if it was generated with the current model or another one). The idea is that the quality of a label set is judged solely by the zero-shot accuracy, so if a label set found with a small model achieves a high zero-shot accuracy with a big model, the learning should behave as described in the paper.
>
> >Q2.  alternate ways of choosing different-quality label sets
>
> This was mostly a design choice motivated by the fact that a user could have access to limited development data on which to fit the labels and we wanted to evaluate the quality of label sets under different data resources. We believe it is also possible to choose earlier solutions in the optimization process although it might converge very quickly which would not result in sufficient sets. Moreover, since it is based on coordinate ascent and we change one token at a time, consecutive label sets would overlap in everything but one token.
>
> >Comment
>
> Lastly, we will make edits such that the flow of some paragraphs is more natural and not simply an enumeration.
>
> Please let us know if our understanding of the review is correct and if our response addresses your concerns! Thank you again for your review!

---

### Author Response · Authors · 2025-11-26
**Revision**

Dear reviewers,

Thank you kindly for your thoughtful feedback! We uploaded a new revision incorporating all the changes you suggested.  Specifically, we added:
1. new experiments with different models (Mistral and Qwen) in Appendix C
2. new experiments with a different dataset (TREC) in Appendix D
3. discussion of limitations in Conclusion
4. implementation details and runtime in Appendix E
5. further explanations of our findings in Results.

Please let us know if your concerns have been addressed. We are looking forward to your responses!

---

### Meta-Review · Area_Chair_Qymj · 2026-01-07

**Summary:**

The paper focuses on the modern topic of in-context learning, which has seen increasing attention over the last two years. The paper introduces an optimization framework to generate label sets of varying semantic quality, aiming to demonstrate that In-Context Learning performance is driven by two orthogonal factors: static representation quality and dynamic learning from demonstrations. The experimental results are mixed and indicate that the rate of learning is functionally dependent on both the representation quality and model scale.

**Reviewer Concerns:**

The reviewers raise multiple concerns, which can be summarized as follows:

1. Narrow scope of the dataset.
2. Limited number of models.
3. Limited number of tasks (thus generalization to general ICL under question).
4. Incomplete experimental details.
5. The main claim of "independence/orthogonality".

Out of those, I believe 4 was addressed during the rebuttal, while for the rest, the rebuttal does provide an answer, but an additional round of discussion would have contributed here.

**Reviewer Scores:**

The original rebuttal was posted on the 17th, so the reviewers did have several days to respond to the original rebuttal. Nevertheless, my guess would be that the scores would not change much in this paper, because many of the complaints were based on the experimental results, which are hard to run in few days with the computationally intensive LLMs (and extend the results beyond Llama).

---

### Decision · Program_Chairs · 2026-01-26

Reject